# ACCELERATING REGRESSION TASKS WITH QUANTUM ALGORITHMS

## ABSTRACT

Regression is a cornerstone of statistics and machine learning, with applications spanning science, engineering, and economics. While quantum algorithms for regression have attracted considerable attention, most existing work has focused on linear regression, leaving many more complex yet practically important variants unexplored. In this work, we present a unified quantum framework for accelerating a broad class of regression tasks—including linear and multiple regression, Lasso, Ridge, Huber, $\ell_p$-, and $\gamma_p$-type regressions—achieving up to a quadratic improvement in the number of samples $m$ over the best classical algorithms. This speedup is achieved by extending the recent classical breakthrough of Jambulapati et al. (2024) using several quantum techniques, including quantum leverage score approximation (Apers & Gribling, 2024) and the preparation of many copies of a quantum state (Hamoudi, 2022). For problems of dimension $n$, sparsity $r < n$, and error parameter $\varepsilon$, our algorithm solves the problem in $\widetilde{O}(r\sqrt{mn}/\varepsilon + \mathrm{poly}(n, 1/\varepsilon))$[1] quantum time, demonstrating both the applicability and the efficiency of quantum computing in accelerating regression tasks.

## 1 INTRODUCTION

Regression lies at the heart of statistical modeling and data analysis, providing a fundamental framework for understanding relationships between variables and making predictions. Among its various forms, linear regression stands out as one of the most widely used and longest-standing tools, with a history dating back over three centuries and applications ranging from the natural sciences to economics and engineering (Legendre, 1806; Fisher, 1992; Neter et al., 1996). Over the decades, the basic linear regression model has been extended in multiple directions, including ridge and lasso regression, $\ell_p$ regression, Huber regression, each addressing different statistical and computational challenges. These models have become indispensable in modern data analysis, driving extensive theoretical research on fast regression methods. Among them, $\ell_p$ regression has witnessed a series of recent breakthroughs (Adil et al., 2019; Lee et al., 2019; Musco et al., 2022; Adil et al., 2024; Jambulapati et al., 2024), alongside many other progress in other regression tasks.

With the rapid progress of quantum hardware and algorithms, there has been growing interest in leveraging quantum computing to accelerate optimization and machine learning tasks. Among them, regression stands out as a core problem in statistics and machine learning, and has attracted significant attention from the quantum algorithms community. A rich body of work has explored quantum approaches to linear regression under various models and assumptions, ranging from HHL-based solvers and conditional gradient method to quantum leverage score techniques (Wiebe et al., 2012; Schuld et al., 2016; Wang, 2017; Kerenidis & Prakash, 2017; Chakraborty et al., 2019; Shao & Xiang, 2020; Chen & de Wolf, 2023; Shao, 2023; Chen et al., 2023; Chakraborty et al., 2023; Song et al., 2023). However, for broader and both theoretically and practically important regression tasks such as $\ell_p$ regression and Huber regression, it remains unclear whether similar quantum speedups can be achieved. Given the diversity of regression tasks and their substantial significance in both machine learning and theoretical computer science, a natural question arises:

---

[1] We use $\widetilde{O}(f)$ to represent $O\left(f \cdot \mathrm{polylog}(m, n, 1/\varepsilon)\right)$ throughout this paper to suppress polylogarithmic factors.

*To what extent can the advantages of quantum computing be extended beyond linear regression to encompass this broader family of models?*

In this work, we study this question by adopting the generalized linear model (GLM) sparsification framework and developing a quantum algorithmic approach tailored to it. Sparsification refers to the process of replacing a large dataset with a much smaller weighted subset (a sparsifier) that approximately preserves the objective value for all candidate solutions, thereby allowing the original optimization problem to be solved more efficiently. Once such a sparsifier is constructed, computations performed on it yield an approximate solution to the original problem. Consequently, our framework provides quantum speedups for approximately solving a broad class of empirical risk minimization (ERM) problems, covering linear and multiple regression, Lasso, ridge, Huber, $\ell_p$-, and $\delta_p$-type regressions. This demonstrates both the unifying nature and the broad applicability of our approach in accelerating fundamental regression tasks.

To clearly specify the scope of our framework, we formalize the problem of optimizing a generalized linear model as

$$\min_{x \in \mathbb{R}^n} F(x) \quad \text{for} \quad F(x) := \sum_{i=1}^m f_i\big(\langle a_i, x\rangle - b_i\big), \tag{1}$$

where the total loss $F : \mathbb{R}^n \to \mathbb{R}$, is defined by data vectors $a_1, \ldots, a_m \in \mathbb{R}^n$, a response vector $b \in \mathbb{R}^m$, and loss functions $f_1, \ldots, f_m : \mathbb{R} \to \mathbb{R}_+$. This general formulation serves as a unifying abstraction for many regression tasks, from linear regression to more sophisticated robust objectives.

When the number of samples $m$ is much larger than the dimension $n$, a natural strategy for accelerating computation is to apply sparsification. This technique has been widely used in classical algorithm design, with graph sparsification being one of the most prominent examples (Spielman & Srivastava, 2011; Batson et al., 2012). In a recent breakthrough, Jambulapati et al. (2024) systematically studied the GLM sparsification problem and presented a corresponding classical algorithm. Motivated by this advance, we investigate the potential for quantum speedups in this setting, leading to our quantum algorithmic GLM sparsification framework. To precisely describe GLM sparsification, we use the following definition:

**Definition 1** (Generalized Linear Model (GLM) Sparsification). *Let the total loss function $F(x) := \sum_{i=1}^m f_i\left(\langle a_i, x\rangle - b_i\right)$ be defined by vectors $a_1, \ldots, a_m \in \mathbb{R}^n$, a vector $b \in \mathbb{R}^m$, and loss functions $f_1, \ldots, f_m : \mathbb{R} \to \mathbb{R}_+$. We say that $F$ admits an $s$-sparse $\varepsilon$-approximation over the range $[s_{\min}, s_{\max}]$ if there exist non-negative weights $w \in \mathbb{R}_+^m$, with at most $s$ nonzero entries, such that*

$$\left| F(x) - \widetilde{F}(x) \right| \le \varepsilon \cdot F(x), \quad \forall x \in \mathbb{R}^n \text{ with } s_{\min} \le F(x) \le s_{\max},$$

*where $\widetilde{F}(x) := \sum_{i=1}^m w_i f_i\left(\langle a_i, x\rangle - b_i\right)$ is referred to as a $s$-sparse $\varepsilon$-approximate sparsifier of $F$ over the range $[s_{\min}, s_{\max}]$.*

## 1.1 OUR RESULTS

We consider a generalized linear model as in Eq. (1), where the loss functions $f_1, \ldots, f_m : \mathbb{R} \to \mathbb{R}_+$ are applied to $\langle a_i, x\rangle - b_i$ for $a_i \in \mathbb{R}^n$. Throughout the analysis, we assume for convenience that the bias term is absent, i.e., $b = 0$. This assumption entails no loss of generality: any nonzero bias $b_i$ can be incorporated into the vector by appending an additional coordinate, rewriting $\langle a_i, x\rangle - b_i$ as $\langle (a_i, b_i), (x, -1)\rangle$, thereby embedding the problem in $\mathbb{R}^{n+1}$ with zero bias.

To enable effective GLM sparsification, we require that each loss function $f_i$ satisfies certain regularity conditions. Specifically, we focus on losses that are $L$-auto-Lipschitz and lower $\theta$-homogeneous—two mild and widely satisfied properties that imply several other desirable characteristics for sparsification. These conditions, along with their consequences, are discussed in detail by Jambulapati et al. (2024). Following their results, we formalize this class of losses as $(L, \theta, c)$-proper functions, defined as follows:

**Definition 2** (Proper Loss Functions). *A family of loss functions $\mathcal{F} = \{f_1, \ldots, f_m\}$ is called $(L, \theta, c)$-proper if for each $f_i : \mathbb{R} \to \mathbb{R}_+$, the transformed function $h_i = \sqrt{f_i}$ satisfies:*

1. *($L$-auto-Lipschitz) $|h_i(x) - h_i(x')| \le L h_i(x - x')$ for all $x, x' \in \mathbb{R}$.*

2. *(Lower $\theta$-homogeneous with $c$) $h_i(\lambda x) \ge c\lambda^\theta h_i(x)$ for all $x \in \mathbb{R}$ and $\lambda \ge 1$.*

We assume $c > 0, L > 0, \theta \in (0, 4)$; in particular, $\theta = 1$ already suffices for all applications in this paper.

In this paper, we introduce a quantum algorithmic framework for GLM sparsification, capable of efficiently constructing $\varepsilon$-approximate sparsifiers for a broad class of $(L, \theta, c)$-proper loss functions. Given query access to the data matrix and the loss functions, our method achieves quadratic speedups in the number of data points $m$ over the fastest classical algorithms, while preserving the same approximation guarantees. Our main result is summarized in the following theorem.

**Theorem 3** (Informal, see Theorem 10 for the formal statement). *Let $F(x) := \sum_{i=1}^{m} f_i(\langle a_i, x \rangle)$ be a total loss function defined by vectors $a_1, \ldots, a_m \in \mathbb{R}^n$ (with sparsity $r \leq n$) and a $(L, \theta, c)$-proper loss functions family $\mathcal{F} = \{f_1, \ldots, f_m\}$. For any $\varepsilon = O(1/r), s_{\max} > s_{\min} \geq 0$, there exists a quantum algorithm that, with high probability[2], constructs an $\widetilde{O}(\log(s_{\max}/s_{\min}) \cdot n/\varepsilon^2)$-sparse $\varepsilon$-approximate sparsifier of $F$ over the range $[s_{\min}, s_{\max}]$, in time $\widetilde{O}\left((r\sqrt{mn}/\varepsilon + \text{poly}(n)) \cdot \log(s_{\max}/s_{\min})\right)$.*

We illustrate applicability with common loss functions. It is straightforward to verify that $\ell_p(x) = |x|^p$ for $p \in (0, 2]$ is $(L, \theta, c)$-proper with $L = 1, \theta = p/2$, and $c = 1$. For the more refined function $\gamma_p$ defined for $p \in (0, 2]$ by

$$\gamma_p(x) := \begin{cases} \frac{p}{2}x^2 & \text{for } |x| \leq 1, \\ |x|^p - (1 - p/2) & \text{otherwise,} \end{cases} \tag{2}$$

one similarly checks $(L, \theta, c)$-properness with the same parameters, where the case $p = 1$ recovers the Huber loss. Regarding the dependence on the scale ratio $s_{\max}/s_{\min}$, when all $f_i$ are $p$-homogeneous the dependence disappears, and for $\gamma_p$ the scale ratio reduces to $\text{poly}(m)$.

Table 1: Comparison of runtimes for various regression problems in classical algorithms, previous quantum algorithms, and our work. Due to space constraints, we present only simplified expressions of the time complexities for each task; detailed discussions are provided in Section 4.2. We list only the state-of-the-art results that allow direct comparison and omit data-dependent algorithms.

| Regression type | Classical | Quantum (previous) | Quantum (this work) |
|---|---|---|---|
| Linear regression | $\widetilde{O}(mr + n^3)$ (Nelson & Nguyễn, 2013; Clarkson & Woodruff, 2017) | $\widetilde{O}(\sqrt{m}n^{1.5}/\varepsilon + \text{poly}(n, 1/\varepsilon))$ (Song et al., 2023) | $\widetilde{O}(r\sqrt{mn}/\varepsilon + n^3)$ (Corollary 25) |
| Multiple regression | $\widetilde{O}(mr + N\,\text{poly}(n, 1/\varepsilon))$ (Nelson & Nguyễn, 2013; Clarkson & Woodruff, 2017) | $\widetilde{O}(\sqrt{m}n^{1.5}/\varepsilon + N\,\text{poly}(n, 1/\varepsilon))$ (Song et al., 2023) | $\widetilde{O}(r\sqrt{mn}/\varepsilon + N\,\text{poly}(n, 1/\varepsilon))$ (Corollary 26) |
| Ridge regression | $\widetilde{O}(mr + \text{poly}(n, 1/\varepsilon))$ (Avron et al., 2017) | $\widetilde{O}(\sqrt{m}n^{1.5}/\varepsilon + \text{poly}(n, 1/\varepsilon))$ (Song et al., 2023) | $\widetilde{O}(r\sqrt{mn}/\varepsilon + n^3)$ (Corollary 27) |
| Lasso regression | $\widetilde{O}(mn^2 + n^3)$ (Efron et al., 2004) | — | $\widetilde{O}(r\sqrt{mn}/\varepsilon + \text{poly}(n, 1/\varepsilon))$ (Corollary 28) |
| Huber regression | $\widetilde{O}(mr + \text{poly}(n, 1/\varepsilon))$ (Jambulapati et al., 2024) | — | $\widetilde{O}(r\sqrt{mn}/\varepsilon + \text{poly}(n, 1/\varepsilon))$ (Corollary 14, with $p = 1$) |
| $\ell_p$ regression | $\widetilde{O}(mr + \text{poly}(n, 1/\varepsilon))$ (Jambulapati et al., 2024) | — | $\widetilde{O}(r\sqrt{mn}/\varepsilon + \text{poly}(n, 1/\varepsilon))$ (Corollary 13) |
| $\gamma_p$ regression | $\widetilde{O}(mr + \text{poly}(n, 1/\varepsilon))$ (Jambulapati et al., 2024) | — | $\widetilde{O}(r\sqrt{mn}/\varepsilon + \text{poly}(n, 1/\varepsilon))$ (Corollary 14) |

As an application of our framework, we achieve quantum speedups for a range of important regression problems, including linear regression, multiple regression, Lasso, Ridge, Huber, $\ell_p$ regression, and $\gamma_p$ regression. Since the sparsifier has size $\widetilde{O}(n/\varepsilon^2)$, it must be smaller than $m$ (otherwise it would be dense), which requires $\varepsilon = \Omega(\sqrt{n/m})$. In this regime, the leading term in our quantum runtime, $r\sqrt{mn}/\varepsilon$, is strictly smaller than the classical $mr$; when $\varepsilon$ is constant, this yields quadratic speedups in the parameter $m$, which often dominates in large-scale regression tasks where $m \gg n$. Moreover, for linear, multiple, and Ridge regression, our algorithm also subsumes the results of Song et al. (2023), the previous known quantum algorithm for the problem.

We emphasize that our focus is on the *computational task* of approximately minimizing the loss function $F$, rather than on the *statistical learning problem* of estimating an unknown model from

---

[2]Throughout this paper, we say something holds "with high probability" if it holds with probability at least $1 - O(1/n)$.

random samples. The latter setting was studied by Chen & de Wolf (2023), who investigated quantum algorithms for Lasso and ridge regression; in our notation, their results correspond to the case $m = O(\log n/\varepsilon^2)$. For clarity, Table 1 summarizes the resulting runtimes across classical algorithms, prior quantum work, and our algorithms.

**Concurrent work**   In a concurrent and independent work, Song et al. (2025) study $\ell_p$ regression within a quantum sensitivity-sampling framework and show that one can construct an $\varepsilon$-coreset of size $\widetilde{O}(n/\varepsilon^2)$ in time $\widetilde{O}(n\sqrt{mn}(\varepsilon^{-3} + \sqrt{n}))$.

## 1.2 TECHNIQUES

**Generalized linear model sparsification.**   To systematically investigate quantum speedups for a broad class of regression problems, we adopt the generalized linear model (GLM) sparsification framework and further extend its applicability by allowing linear combinations of loss functions. At a high level, GLM sparsification replaces a large dataset with a much smaller weighted subset (the sparsifier) such that the total loss is preserved within a $(1 \pm \varepsilon)$ factor for all choices of the variable $x$. This enables the original optimization problem to be approximated efficiently by solving it on the sparsifier. Recently established in the breakthrough works of Jambulapati et al. (2023; 2024), this framework unifies and generalizes sparsification techniques for a wide range of regression objectives. Its core idea is to compute, for each loss function $f_i$, an importance score capturing its contribution to the objective, and then perform importance sampling to select a small representative set, reweighting them to form the sparsifier. In the quantum setting, our approach comprises two parts: (i) computing importance scores efficiently, and (ii) performing importance sampling based on these scores.

**Multiscale leverage score overestimates.**   Defining suitable importance scores in the GLM setting is challenging due to its high level of abstraction: rather than specifying a concrete loss function, only general conditions on $f_i$ are given. Jambulapati et al. (2024) addressed this by analyzing the iterated covering argument for $\ell_p$ loss and generalizing $\ell_p$ Lewis weights (Lewis, 1979; Bourgain et al., 1989; Cohen & Peng, 2015), yielding an *approximate weight scheme* consisting of multiple sets of weights. These can be interpreted as importance scores at different scales, connected by specific contraction properties. Building on this, we summarized the key structural ideas into the notion of *multiscale leverage score overestimates* (MLSO), which serves as the GLM analogue of importance scores.

To achieve quantum speedups, we use the results of Apers & Gribling (2024) on quantum leverage score approximation as a key foundation, enabling a quantum implementation of the contractive algorithm from Jambulapati et al. (2024). This yields a quantum data structure, built with sublinear-time preprocessing, that supports efficient queries to MLSO values. Conceptually, our algorithm iteratively refines an initial coarse weight estimate over $\widetilde{O}(1)$ rounds, followed by a final quantum refinement step to produce MLSO values. To support this process, we develop a quantum procedure for constructing the initial weight vector by slightly modifying each $f_i$ and generating an appropriate starting weight, ensuring that the entire pipeline remains fully quantum.

**Quantum importance sampling.**   Once quantum query access to the importance scores is available, the next step is to perform importance sampling efficiently. The main challenge is that we do not have direct access to the normalized sampling distribution, but only to an oracle returning unnormalized importance scores. To address this, we use the "preparing many copies of a quantum state" technique of Hamoudi (2022), which allows us to prepare many copies of a quantum state whose amplitudes are proportional to the importance scores. This technique is a powerful and widely applicable tool, and has also been used in several other efficient quantum algorithms (see, e.g., Xue et al., 2023; Wang et al., 2024; Liu et al., 2025; Apers et al., 2025). We combine this with a quantum sum estimation procedure (Li et al., 2019) to approximate the normalization constant, enabling correct reweighting of the sampled elements. The correctness of this method follows from a careful adaptation of the guarantees established in Jambulapati et al. (2024).

## 2 PRELIMINARIES

**Notations.** For a positive integer $n \in \mathbb{Z}_+$, we use $[n]$ to denote the set $1, 2, \ldots, n$ and $[n]_0$ to denote the set $0, 1, \ldots, n-1$. For vectors $u, v \in \mathbb{R}^n$, we write $\langle u, v \rangle = \sum_{i \in [n]} u_i v_i$ for the standard inner product. For a vector $w \in \mathbb{R}^n$, we use $\mathrm{diag}(w)$ to denote the diagonal $n \times n$ matrix whose $i$-th diagonal entry equals the $i$-th element of $w$.

**Quantum computational model.** We assume the usual quantum computational model, which is a classical system that can (i) run quantum subroutines on $O(\log m)$ qubits, (ii) can make quantum queries to the input, and (iii) has access to a quantum-read/classical write RAM (QRAM) of $\mathrm{poly}(m)$ bits. The *quantum query complexity* measures the number of quantum queries that an algorithm makes. The *quantum time complexity* measures the number of elementary classical and quantum gates, quantum queries, and single-bit QRAM operations that algorithm uses. For readers less familiar with the basics of quantum computing, we refer to Appendix A.

Regarding queries to the input, we consider an input matrix $A \in \mathbb{R}^{m \times n}$ with row sparsity $r \leq n$, where the matrix elements can be queried via the oracle $\mathcal{O}_A^{\mathrm{elem}} : |i\rangle|j\rangle|0\rangle \mapsto |i\rangle|j\rangle|a_{ij}\rangle, \forall i \in [m], j \in [n]$. In addition, the indices of the nonzero entries of each row can be queried via the oracle $\mathcal{O}_A^{\mathrm{idx}} : |i\rangle|k\rangle \mapsto |i\rangle|t_{ik}\rangle, \forall i \in [m], k \in [r_i]$, where $t_{ik}$ is the column index of the $k$-th nonzero entry in the $i$-th row of $A$, and $r_i \leq r$ is the number of nonzero entries in that row. For simplicity, we use the notation $\mathcal{O}_A$ to refer to either of these two oracles. We will also use the following quantum queries: for a vector $w \in \mathbb{R}^n$, its elements can be queried via the oracle $\mathcal{O}_w : |i\rangle|0\rangle \mapsto |i\rangle|w_i\rangle, i \in [n]$; and for a family of functions $\mathcal{F} = \{f_1, \ldots, f_m\}$, we assume access to the oracle $\mathcal{O}_{\mathcal{F}} : |i\rangle|x\rangle|0\rangle \mapsto |i\rangle|x\rangle|f_i(x)\rangle$.

**Quantum leverage score approximation.** Let $A \in \mathbb{R}^{m \times n}$ be a matrix with rows $a_1, \ldots, a_m$. The *leverage score* of the $i$-th row is defined as

$$\sigma_i(A) := a_i^\top (A^\top A)^+ a_i, \quad \forall i \in [m], \tag{3}$$

where $(A^\top A)^+$ denotes the Moore-Penrose pseudoinverse of $A^\top A$. Intuitively, $\sigma_i(A)$ measures the "statistical importance" of row $i$ with respect to the whole set of rows. Leverage scores are central to row-sampling methods that construct a smaller matrix $\widetilde{A}$ preserving the key spectral properties of $A$. Sampling rows proportionally to $\sigma_i(A)$ and rescaling yields $\widetilde{A}^\top \widetilde{A} \approx A^\top A$ with high probability, enabling accurate and efficient approximation of linear regression and other optimization tasks; we refer the reader to the survey of Woodruff (2014) for a comprehensive overview.

In a recent important work, Apers & Gribling (2024) introduced a simple recursive halving algorithm for quantum spectral approximation. As a corollary, they obtained a quantum algorithm for leverage score approximation, which serves as a cornerstone of our work—we invoke this result multiple times throughout our algorithms. For completeness, we restate their result below.

**Theorem 4** (Quantum Leverage Score Approximation). *(Apers & Gribling, 2024, Theorem 3.2) Assume query access to a matrix $A = \mathbb{R}^{m \times n}$ with row sparsity $r$. For any $0 < \varepsilon \leq 1$, there exists a quantum algorithm that, with high probability, provides quantum query access to estimates $\widetilde{\sigma}_i$ for any $i \in [m]$ satisfying $(1 - \varepsilon)\sigma_i(A) \leq \widetilde{\sigma}_i \leq (1 + \varepsilon)\sigma_i(A)$. The algorithm makes $\widetilde{O}(\sqrt{mn}/\varepsilon)$ row queries to $A$ and runs in $\widetilde{O}(r\sqrt{mn}/\varepsilon + n^\omega + \min\{n^\omega, nr^2\}/\varepsilon^2 + n^2/\varepsilon^4)$ time. Furthermore, the cost to estimate per $\widetilde{\sigma}_i$ is one row query to $A$ and $\widetilde{O}(r/\varepsilon^2)$ time.*

We note that a row query to the $i$-th row of $A$ returns the entire row, and the query can be made in superposition. Equivalently, one row query corresponds to $O(r)$ queries to $\mathcal{O}_A$, since each row contains at most $r$ nonzero entries.

## 3 QUANTUM ALGORITHM FOR MULTISCALE LEVERAGE SCORE OVERESTIMATES

In this section, we present our core quantum algorithm, *Quantum Multiscale Leverage Score Overestimates* (QMLSO), the key to enabling quantum acceleration in our GLM sparsification framework. We begin by defining MLSO, distilling the structural ideas from the approximate weight scheme of

Jambulapati et al. (2024), which serves as the GLM analogue of importance scores. We then describe our quantum procedure for computing MLSO values from an initial coarse weight estimate, using quantum leverage score approximation to achieve sublinear-time refinement. Finally, we develop a quantum algorithm for constructing suitable initial weights, ensuring that the entire MLSO computation can be executed fully within the quantum setting.

## 3.1 MULTISCALE LEVERAGE SCORE OVERESTIMATES (MLSO)

We begin by recalling the definition of approximate weights introduced by Jambulapati et al. (2024), which generalizes the notion of $\ell_p$ Lewis weights to capture the statistical contribution of each row at a given scale, where the scale refers to the magnitude range of residuals being considered.

**Definition 5** (Approximate Weights; see Jambulapati et al. (2024, Definition 1.7)). *Fix a matrix $A \in \mathbb{R}^{m \times n}$ (where the $i$-th row is $a_i$ for each $i \in [m]$) and a family of loss functions $\mathcal{F} = \{f_1, \ldots, f_m\}$. We say a vector $w \in \mathbb{R}_+^m$ is an $\alpha$-approximate weight (with respect to $A$ and $\mathcal{F}$) at scale $s$ if, for all $i \in [m]$,*

$$\frac{s}{\alpha} \leq \frac{f_i(\|M^{-1/2}a_i\|_2)}{w_i\|M^{-1/2}a_i\|_2^2} \leq \alpha s, \quad where \quad M := \sum_{i=1}^m w_i a_i a_i^\top.$$

Since approximate weights are defined only at a single scale, they are not sufficient for handling general loss functions where homogeneity fails. To address this, Jambulapati et al. (2024) introduced weight schemes—collections of approximate weights defined consistently across a range of scales. These schemes guarantee that the contribution of each row is well controlled not only at a single scale but simultaneously across multiple scales, thereby capturing the multiscale structure inherent in $A$ and $\mathcal{F}$.

**Definition 6** (Weight Schemes; see Jambulapati et al. (2024, Definition 1.8)). *Fix a matrix $A \in \mathbb{R}^{m \times n}$ (the $i$-th row is $a_i$ for each $i \in [m]$) and a family of loss functions $\mathcal{F} = \{f_1, \ldots, f_m\}$. Let $\mathcal{J} \subseteq \mathbb{Z}$ be a contiguous interval. We say that a family $\mathcal{W}_{A,\mathcal{F},\mathcal{J}} = \{w^{(j)} \in \mathbb{R}_+^m : j \in \mathcal{J}\}$ is an $\alpha$-approximate weight scheme if:*

*1. Each $w^{(j)}$ is an $\alpha$-approximate weight (with respect to $A$ and $f$) at scale $2^j$;*

*2. For every pair $j, j+1 \in \mathcal{J}$, we have $w_i^{(j+1)} \leq \alpha w_i^{(j)}, \forall i \in [m]$.*

*We omit the subscript $(A, \mathcal{F}, \mathcal{J})$ when it is clear from context.*

When working with weight schemes, we ultimately need a compact summary that upper bounds the statistical contribution of each row across all relevant scales. This motivates the notion of multiscale leverage score overestimates. By aggregating leverage scores over the entire weight scheme, such overestimates provide a single vector that controls row contributions across all scales. This will later serve as the key object enabling efficient sampling and sparsification in our quantum algorithm.

**Definition 7** (Multiscale Leverage Score Overestimates). *Let $A \in \mathbb{R}^{m \times n}$ be a matrix with rows $a_1, \ldots, a_m$, let $\mathcal{F} = \{f_1, \ldots, f_m\}$ be a family of loss functions, and let $\mathcal{J} \subseteq \mathbb{Z}$ be a contiguous interval. Suppose $\mathcal{W} = \{w^{(j)} \in \mathbb{R}_+^m : j \in \mathcal{J}\}$ is an $\alpha$-approximate weight scheme with respect to $A, \mathcal{F}, \mathcal{J}$. We say that a vector $z \in \mathbb{R}_+^m$ is a multiscale (leverage score) $\tau$-overestimates with respect to $A$ and $\mathcal{W}$ if*

$$\|z\|_1 \leq \tau, \quad and \quad z_i \geq \sigma_i^{\mathrm{ms}}(A, \mathcal{W}), \quad \forall i \in [m],$$

*where the multiscale leverage score of the $i$-th row is defined as*

$$\sigma_i^{\mathrm{ms}}(A, \mathcal{W}) := \max_{j \in \mathcal{J}} \sigma_i(W_j^{1/2}A), \quad \forall i \in [m],$$

*and $W_j := \mathrm{diag}(w^{(j)})$ for each $j \in \mathcal{J}$.*

## 3.2 QUANTUM ALGORITHM FOR MLSO

We now present our quantum algorithm for computing multiscale leverage score overestimates (MLSO), which serves as a key subroutine in our main quantum algorithm for GLM sparsification.

Our approach builds upon the classical contractive algorithm of Cohen & Peng (2015); Jambulapati et al. (2024), but is carefully adapted to the quantum setting.

The input consists of quantum oracles for the matrix $A$ and the function family $\mathcal{F}$, along with query access to an initial approximate weight vector $w^\circ$ and relevant parameters. The output is a data structure that provides efficient quantum query access to multiscale leverage score overestimates (see Proposition 17 for a formal description).

The algorithm proceeds iteratively: over $\widetilde{O}(1)$ rounds, it refines the largest-scale weight until it becomes well-conditioned. Once this condition is achieved, the algorithm recursively computes the weights at smaller scales and combines them to obtain the final MLSO. Within each iteration, the subroutine ModLevApprox supplies quantum query access to the modified leverage scores (Proposition 15), while WeightCompute produces quantum query access to the updated weights for the subsequent round (Proposition 16). The complete procedure is summarized in Algorithm 1, and its formal guarantees are established in Theorem 8. Owing to space limitations, the proof of Theorem 8 is deferred to Appendix B.

---

**Algorithm 1** QMLSO$(\mathcal{O}_\mathcal{F}, \mathcal{O}_A, \mathcal{O}_{w^\circ}, j_{\min}, j_{\max}, \beta, L, \theta, c)$

---

**Require:** A matrix oracle $\mathcal{O}_A$; a proper loss functions oracle $\mathcal{O}_\mathcal{F}$ with parameters $L, \theta, c$; an initial weight oracle $\mathcal{O}_{w^\circ}$ with associated parameters $\beta$ and $j_{\max}$; and an integer $j_{\min} < j_{\max}$.

**Ensure:** An instance $\mathcal{Z}$ of QOverestimate which stores the vector $z$ being a multiscale $O(n \cdot |j_{\max} - j_{\min}|)$-overestimate with respect to matrix $A$ and $\alpha$-approximate scheme $\mathcal{W} = \{w^{(j)} \in \mathbb{R}_+^m : j \in \mathcal{J} := \mathbb{Z} \cap [j_{\min}, j_{\max}]\}$ for some $\alpha = \alpha(L, \theta, c)$.

1: $\varepsilon \leftarrow 0.1$, $\delta \leftarrow \max\left\{\frac{1}{2}, \left|\frac{\theta-2}{2}\right|\right\}$, $C \leftarrow \max\left\{\frac{2L}{c}, \frac{1}{c}\right\}$

2: $T \leftarrow \left\lceil \left(\log\log(\frac{1-\varepsilon}{(1+\varepsilon)^2} \cdot C) - \log\log\beta\right) / \log\delta \right\rceil$

3: $U_{w_{\text{itr}}^{(0)}} \leftarrow \mathcal{O}_{w^\circ}$

4: **for** $i = 0$ to $T$ **do**

5: $\quad \mathcal{L}_{\text{itr}}^{(i)} \leftarrow$ ModLevApprox$(\mathcal{O}_A, U_{w_{\text{itr}}^{(i)}}, \varepsilon)$

6: $\quad U_{w_{\text{itr}}^{(i+1)}} \leftarrow$ WeightCompute$(\mathcal{O}_\mathcal{F}, U_{w_{\text{itr}}^{(i)}}, \mathcal{L}_{\text{itr}}^{(i)}, 2^{j_{\max}})$

7: **end for**

8: $U_{w^{(j_{\max})}} \leftarrow U_{w_{\text{itr}}^{(T)}}$

9: **for** $j = j_{\max} - 1, \ldots$ to $j_{\min}$ **do**

10: $\quad \mathcal{L}^{(j+1)} \leftarrow$ ModLevApprox$(\mathcal{O}_A, U_{w^{(j+1)}}, \varepsilon)$

11: $\quad U_{w^{(j)}} \leftarrow$ WeightCompute$(\mathcal{O}_\mathcal{F}, U_{w^{(j+1)}}, \mathcal{L}^{(j+1)}, 2^j)$

12: **end for**

13: $\mathcal{Z} \leftarrow$ QOverestimate$(\{\mathcal{L}^{(j)} : j \in \mathcal{J}\}, \varepsilon)$

---

**Theorem 8** (Quantum Multiscale Leverage Score Overestimates)**.** *There is a quantum algorithm* QMLSO$(\mathcal{O}_\mathcal{F}, \mathcal{O}_A, \mathcal{O}_{w^\circ}, j_{\min}, j_{\max}, \beta, L, \theta, c)$ *that, given query access $\mathcal{O}_A$ to a matrix $A = \mathbb{R}^{m \times n}$ with row sparsity $r \leq n$, query access $\mathcal{O}_\mathcal{F}$ to a $(L, \theta, c)$-proper loss functions family $\mathcal{F} = \{f_1, \ldots, f_m\}$, query access $\mathcal{O}_{w^\circ}$ to a initial weight $w^\circ \in \mathbb{R}_+^m$ which is a $\beta$-approximate weight at scale $2^{j_{\max}}$ (with respect to $A$ and $\mathcal{F}$) for some integer $j_{\max}$, and any integer $j_{\min} < j_{\max}$, the algorithm makes $O(\sqrt{mn})$ queries to $\mathcal{O}_{w^\circ}$, $O(r\sqrt{mn}\Delta)$ queries to $\mathcal{O}_A$, $O(\sqrt{mn}\Delta)$ queries to $\mathcal{O}_\mathcal{F}$, runs in time $\widetilde{O}\left((r\sqrt{mn} + n^\omega + nr^2)\Delta\right)$, where $\Delta = |j_{\max} - j_{\min}| + \max\{\log\log\beta, 0\}$. Then, with high probability, it provides query access to a multiscale $O(n \cdot |j_{\max} - j_{\min}|)$-overestimates $z$ with respect to $A$ and $\alpha$-approximate weight scheme $\mathcal{W} = \{w^{(j)} \in \mathbb{R}_+^m : j \in \mathcal{J} := \mathbb{Z} \cap [j_{\min}, j_{\max}]\}$ for some $\alpha = \alpha(L, \theta, c)$, where each query to $z$ requires $\widetilde{O}(r|\mathcal{J}|)$ time.*

## 3.3 INITIAL WEIGHT CONSTRUCTION

When applying our QMLSO algorithm, it is crucial to start with a well-conditioned $\beta$-approximate weight, where $\beta$ cannot be too large. In this subsection, we show how to construct such an initial weight by leveraging quantum leverage score approximation together with a binary search procedure, combined with a mild modification of the loss functions $f_i$. Our initialization algorithm outputs a quantum data structure that, given query access to $A$ and $\mathcal{F}$, as well as the largest scale $s_{\max}$ and a

parameter $\delta$, provides quantum query access to an initial weight vector $w^\circ$ (at scale $s_{\max}$) together with the corresponding modified loss family $\mathcal{F}^\circ$. This construction can be viewed as the quantum analogue of the initialization procedure introduced by Jambulapati et al. (2024). We formalize this construction in the following theorem, and defer its proof to Appendix C.

**Theorem 9** (Weight Initialization). *Assume query access $\mathcal{O}_A$ to a matrix $A = \mathbb{R}^{m \times n}$ with row sparsity $r$ and query access $\mathcal{O}_{\mathcal{F}}$ to a $(L, \theta, c)$-proper loss functions family $\mathcal{F} = \{f_1, \ldots, f_m\}$. There exists a quantum data structure* WeightInitialize *that supports the following operations:*

1. *Initialization:* WeightInitialize$(\mathcal{O}_A, \mathcal{O}_{\mathcal{F}}, s_{\max}, \delta)$, *outputs an instance $\mathcal{A}$, making $\widetilde{O}(r\sqrt{mn})$ queries to $\mathcal{O}_A$ and $\widetilde{O}(\sqrt{mn})$ queries to $\mathcal{O}_{\mathcal{F}}$, in $\widetilde{O}(r\sqrt{mn} + n^\omega + nr^2)$ time.*

2. *Function query: $\mathcal{A}$.QueryFunction, provides a query access $\mathcal{O}_{\mathcal{F}^\circ}$ to a $(\max\{1, L\}, \theta, c)$-proper loss functions family $\mathcal{F}^\circ = \{f_1^\circ, \ldots, f_m^\circ\}$ which satisfies*

$$\sum_{i=1}^m f_i\left(\langle a_i, x\rangle\right) \leq s_{\max} \quad \Rightarrow \quad 0 \leq \sum_{i=1}^m f_i^\circ\left(\langle a_i, x\rangle\right) - f_i\left(\langle a_i, x\rangle\right) \leq C_{\mathrm{INIT}}\delta m^2 s_{\max}, \quad (4)$$

*for some constant $C_{\mathrm{INIT}} = C_{\mathrm{INIT}}(L, \theta, c)$. Each query requires $O(r)$ queries to $\mathcal{O}_A$ and $O(1)$ queries to $\mathcal{O}_{\mathcal{F}}$, and runs in $\widetilde{O}(r)$ time.*

3. *Weight query: $\mathcal{A}$.QueryWeight, provides a query access $\mathcal{O}_{w^\circ}$ to a weight $w^\circ \in \mathbb{R}_+^m$ which is a $\beta$-approximate weight at scale $s_{\max}$ with respect to $A$ and $\mathcal{F}^\circ$, where $\beta = O((L/c)^2 m/\delta)$. Each query requires $O(r)$ queries to $\mathcal{O}_A$ and $O(1)$ queries to $\mathcal{O}_{\mathcal{F}}$, and runs in $\widetilde{O}(r)$ time.*

# 4 QUANTUM ALGORITHM FOR GENERALIZED LINEAR MODELS SPARSIFICATION

In this section, we present our *Quantum Algorithm for GLM Sparsification*, which builds on the MLSO procedure to achieve substantial quantum speedups. The first part outlines our quantum sparsification framework, which implements efficient importance sampling by combining the preparation of many copies of a quantum state (Hamoudi, 2022) with quantum sum estimation (Li et al., 2019), enabling sublinear-time sampling and reweighting. The second part demonstrates the versatility of this framework by embedding a broad class of optimization problems into it, showing that they can be solved with quantum speedups.

## 4.1 QUANTUM ALGORITHM FOR GLM SPARSIFICATION

To describe our quantum sparsification framework, we rely on two fundamental primitives. The first is Multiple Quantum State Preparation, which efficiently produces many copies of a quantum state; measuring these copies yields samples where each index $i$ is selected with probability proportional to $w_i$, thereby enabling importance sampling with sublinear overhead (restated in Theorem 22). The second is Quantum Sum Estimation, which offers a quadratic speedup over classical approaches and is crucial for efficient reweighting in our framework (restated in Theorem 23).

With these tools in place, we now present our main result: the Quantum Algorithm for GLM Sparsification. Building on the QMLSO procedure and the primitives above, our algorithm efficiently sparsifies generalized linear models while achieving polynomial quantum speedups. The algorithm and its formal guarantee are stated below, with the proof deferred to Appendix D.

**Theorem 10** (Quantum Algorithm for GLM Sparsification). *Let $F(x) := \sum_{i=1}^m f_i\left(\langle a_i, x\rangle\right)$ be a total loss function defined by vectors $a_1, \ldots, a_m \in \mathbb{R}^n$ (with sparsity $r \leq n$) and a $(L, \theta, c)$-proper loss functions family $\mathcal{F} = \{f_1, \ldots, f_m\}$. There exists a quantum algorithm* QGLMSparsify$(\mathcal{O}_A, \mathcal{O}_{\mathcal{F}}, L, \theta, c, \varepsilon, s_{\min}, s_{\max})$ *that, given query access $\mathcal{O}_A$ to the matrix $A$ (where the $i$-row is $a_i$), query access $\mathcal{O}_{\mathcal{F}}$ to the function family $\mathcal{F}$ with parameters $L, \theta, c$, and numbers $\varepsilon > 0$, $s_{\max} > s_{\min} \geq 0$, outputs with high probability a non-negative weight vector $w \in \mathbb{R}_+^m$ such that $\widetilde{F} := \sum_{i=1}^m w_i f_i(\langle a_i, x\rangle)$ is an $\widetilde{O}(\log(s_{\max}/s_{\min}) \cdot n/\varepsilon^2)$-sparse $\varepsilon$-approximate sparsifier of $F$ over the range $[s_{\min}, s_{\max}]$, in time $\widetilde{O}\left((n^\omega + nr^2 + r\sqrt{mn}/\varepsilon) \cdot \log(s_{\max}/s_{\min})\right)$.*

---

**Algorithm 2** Quantum GLM Sparsification QGLMSparsify$(\mathcal{O}_\mathcal{F}, \mathcal{O}_A, L, \theta, c, \varepsilon, s_{\min}, s_{\max})$

---

**Require:** A matrix oracle $\mathcal{O}_A$; a proper loss functions oracle $\mathcal{O}_\mathcal{F}$ with parameters $L, \theta, c$; parameters $\varepsilon > 0$, $s_{\max} > s_{\min} \geq 0$.

**Ensure:** A non-negative weight vector $w \in \mathbb{R}_+^m$ such that $\widetilde{F}(x) := \sum_{i=1}^m w_i f_i(\langle a_i, x \rangle)$ is an $\widetilde{O}(\log(s_{\max}/s_{\min}) \cdot n/\varepsilon^2)$-sparse $\varepsilon$-approximate sparsifier of $F$ over the range $[s_{\min}, s_{\max}]$.

1: $\widetilde{w} = 0, t \leftarrow 1, M \leftarrow \widetilde{\Theta}(n/\varepsilon^2), \delta \leftarrow O(\varepsilon s_{\min}/(m^3 s_{\max}))$
2: $j_{\min} \leftarrow \lfloor \log s_{\min} - 4 \log m \rfloor, j_{\max} \leftarrow \lceil \log s_{\max} \rceil, \beta \leftarrow O((L/c)^2 m/\delta)$
3: $\mathcal{A} \leftarrow \mathsf{WeightInitialize}(\mathcal{O}_A, \mathcal{O}_\mathcal{F}, s_{\max}, \delta)$              (Theorem 9)
4: $\mathcal{Z} \leftarrow \mathsf{QMLSO}(\mathcal{A}.\mathsf{QueryFunction}, \mathcal{A}.\mathsf{QueryWeight}, j_{\min}, j_{\max}, \beta, L, \theta, c)$   (Theorem 8)
5: $\vartheta \leftarrow \mathsf{MultiSample}(\mathcal{Z}, M)$                                      (Theorem 22)
6: $\widetilde{\nu} \leftarrow \mathsf{SumEstimate}(\mathcal{Z}, 0.1)$                                 (Theorem 23)
7: **for** $i = 1$ to $M$ **do**
8:     Get $z_{\vartheta_i}$ from $\mathcal{Z}$
9:     $\widetilde{w}_{\vartheta_i} \leftarrow \widetilde{w}_{\vartheta_i} + \widetilde{\nu}/(1.1\, M z_{\vartheta_i})$
10: **end for**
11: Return $\widetilde{w}$.

---

We note that scale ratio term $\log(s_{\max}/s_{\min})$ can be removed when all $f_i$ are $p$-homogeneous for some $p > 0$ (i.e., $f_i(\lambda x) = |\lambda|^p f_i(x)$, and in particular $\ell_p$ is $p$-homogeneous). For non-homogeneous cases such as $\gamma_p$, this can be reduced to $\log(\mathrm{poly}(m))$ via properness assumption (Jambulapati et al., 2024, Lemma 1.5).

**Remark 11.** *In the special case of linear regression, the GLM sparsification framework only requires standard leverage-score overestimates rather than the multiscale leverage score overestimate (MLSO) machinery. Concretely, in Algorithm 2, Steps 2-4 can be replaced by the quantum leverage score routine of Apers & Gribling (2024); under this simplification, Algorithm 2 essentially reduces to a quantum spectral approximation procedure in the sense of Theorem3.1 of Apers & Gribling (2024), and once a spectral sparsifier has been constructed, one can approximately solve linear regression and its simple variants (e.g., multiple and ridge regression) on top of that sparsifier using standard techniques. For more general homogeneous loss functions such as $\ell_p$, the MLSO construction can likewise be reduced to Lewis weights, and Algorithm 1 can be viewed as a quantum iterative approach for producing approximate Lewis weights from approximate leverage scores; related quantum procedures for approximating Lewis weights have also been studied in Apers & Gribling (2024) and in the concurrent work on sublinear-time quantum sensitivity sampling of Song et al. (2025).*

**Remark 12.** *We briefly record two complementary quantum lower bounds that apply to our setting. Since linear regression is a special case of GLM sparsification (by taking $f_i = \ell_2$ for all $i$), any lower bound for quantum spectral approximation immediately yields a lower bound for quantum GLM sparsification. The first lower bound shows the optimality of the $m$-dependence for constant $\varepsilon$: for arbitrary $m \geq n$, there exists a family of matrices $A \in \mathbb{R}^{m \times n}$ for which any quantum algorithm that computes a constant-factor approximation of $F$ must make $\Omega(\sqrt{mn})$ row queries to $A$ (Apers & Gribling, 2024, Theorem8.1). The second lower bound shows optimality of the $\varepsilon$-dependence, but it is restricted to $m \in O(n^2)$: for arbitrary $n$ and $m \in O(n^2)$ and any $\varepsilon \in \Omega(\sqrt{m/n})$, there exists a family of matrices $A \in \mathbb{R}^{m \times n}$ for which any quantum algorithm that computes an $\varepsilon$-approximation of $F$ must make $\Omega(\sqrt{mn}/\varepsilon)$ row queries to $A$ (Apers & de Wolf, 2022, Theorem 2).*

## 4.2 APPLICATIONS

Firstly, it's not hard to verify that both $\ell_p(x) = |x|^p$ and $\gamma_p$ (as defined in Eq. (2)) are $(L, \theta, c)$-proper for $p \in (0, 2]$, with parameters $L = 1$, $\theta = \frac{p}{2}$, and $c = 1$. Applying Theorem 10, this yields a $\widetilde{O}(n/\varepsilon^2)$-sparse $\varepsilon$-approximate sparsifier for the $\ell_p$ and $\gamma_p$ losses, computable in time $\widetilde{O}\left(r\sqrt{mn}/\varepsilon + n^\omega + nr^2\right)$. Given such an $\varepsilon$-approximate sparsifier, one can then invoke a fast classical algorithm to obtain an $\varepsilon$-accurate solution in $\mathrm{poly}(n, 1/\varepsilon)$ time. Notably, the case $p = 2$ for $\ell_p$ recovers linear regression (Corollary 25), while $p = 1$ for $\gamma_p$ corresponds to Huber regression.

**Corollary 13** (Quantum $\ell_p$ Regression)**.** *Let $p \in (0, 2]$. There exists a quantum algorithm that, given query access to a matrix $A \in \mathbb{R}^{m \times n}$ (with row sparsity $r \le n$) and vector $b \in \mathbb{R}^m$, and $\varepsilon > 0$, outputs with high probability an $x \in \mathbb{R}^n$ such that $\|Ax - b\|_p^p \le (1 + \varepsilon) \min_{x \in \mathbb{R}^n} \|Ax - b\|_p^p$, in $\widetilde{O}(r\sqrt{mn}/\varepsilon) + \mathrm{poly}(n, \varepsilon)$ time.*

**Corollary 14** (Quantum $\gamma_p$ Regression)**.** *Let $p \in (0, 2]$. There exists a quantum algorithm that, given query access to a matrix $A \in \mathbb{R}^{m \times n}$ where the $i$-th row is $a_i$ (with sparsity $r \le n$) and vector $b \in \mathbb{R}^m$, and $\varepsilon > 0$, outputs with high probability an $x \in \mathbb{R}^n$ such that $F(x) \le (1 + \varepsilon) \min_{x \in \mathbb{R}^n} F(x)$ for $F(x) = \sum_{i=1}^m \gamma_p(\langle a_i, x \rangle - b_i)$, in $\widetilde{O}(r\sqrt{mn}/\varepsilon) + \mathrm{poly}(n, \varepsilon)$ time.*

Now we go further and consider the case of multiple regression. Given $A \in \mathbb{R}^{m \times n}$ and $B \in \mathbb{R}^{m \times N}$, the goal is to find $X \in \mathbb{R}^{n \times N}$ such that $\|AX - B\|_F \le (1 + \varepsilon) \|AX' - B\|_F$. Writing $X = [x^1, \ldots, x^N]$ and $B = [b^1, \ldots, b^N]$, we expand $\|AX - B\|_F^2 = \sum_{k=1}^N \|Ax^k - b^k\|^2 = \sum_{k=1}^N \sum_{j=1}^m (\langle a_i, x^k \rangle - B_{jk})^2$. Thus, multiple regression reduces to a single large-scale linear regression problem, with an effective data dimension $n' = nN$. See Corollary 26 for the formal description of the quantum algorithm.

We next turn to the ridge regression objective $F_{\mathrm{ridge}}(x) = \|Ax - b\|_2^2 + \lambda \|x\|_2^2$. The regularization term can be written as a sum of coordinate-wise quadratic losses: $\lambda \|x\|_2^2 = \sum_{j=1}^n \left( \sqrt{\lambda} \langle e_j, x \rangle \right)^2$. For using the GLM sparsification framework, we define

$$A' = \begin{bmatrix} A \\ \sqrt{\lambda} I_n \end{bmatrix} \in \mathbb{R}^{(m+n) \times n}, \qquad b' = \begin{bmatrix} b \\ 0_n \end{bmatrix} \in \mathbb{R}^{m+n}, \tag{5}$$

and use the quadratic loss $f(x) = x^2$. Then for every $x \in \mathbb{R}^n$ we have $\sum_{i=1}^{m+n} f\left( \langle a_i', x \rangle - b_i' \right) = \|Ax - b\|_2^2 + \lambda \|x\|_2^2 = F_{\mathrm{ridge}}(x)$. Hence, ridge regression reduces to linear regression with an effective data size $m' = m + n$. The formal quantum algorithm is given in Corollary 27.

Beyond ridge, the lasso regression objective introduces $\ell_1$ regularization: $F_{\mathrm{lasso}}(x) = |Ax - b|_2^2 + \lambda |x|_1$. We define the loss functions $\mathcal{F}' = \{f_1'(x) = \cdots = f_m'(x) = x^2; f_{m+1}'(x) = \cdots = f_{m+n}'(x) = |x|_1\}$, and adopt the notation from Eq. (5). It follows that $\sum_{i=1}^{m+n} f'(\langle a_i', x \rangle - b_i') = \|Ax - b\|_2^2 + \lambda \|x\|_1$. Hence, the GLM sparsification framework applies. The corresponding quantum algorithm is formally described in Corollary 28.

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

## A QUANTUM COMPUTING

In quantum mechanics, a $d$-dimensional quantum state $|v\rangle = (v_0, \ldots, v_{d-1})^\top$ is represented as a unit vector in the Hilbert space $\mathbb{C}^d$, satisfying $\sum_{i \in [d]_0} |v_i|^2 = 1$. The computational basis of this space is $\{|i\rangle\}_{i \in [d]_0}$, where $|i\rangle = (0, \ldots, 0, 1, 0, \ldots, 0)^\top$ with the $i$-th coordinate (0-indexed) equal to 1 and all others 0. For two states $|u\rangle, |v\rangle \in \mathbb{C}^d$, their inner product is $\langle u|v\rangle = \sum_{i \in [d]_0} u_i^* v_i$, with $z^*$ denoting complex conjugation. The tensor product of $|u\rangle \in \mathbb{C}^{d_1}$ and $|v\rangle \in \mathbb{C}^{d_2}$ is their Kronecker product, $|u\rangle \otimes |v\rangle = (u_0 v_0, u_0 v_1, \ldots, u_{d_1-1} v_{d_2-1})^\top$, often abbreviated as $|u\rangle|v\rangle$.

A quantum bit, or qubit, is simply a quantum state in $\mathbb{C}^2$, expressible as $|\psi\rangle = \alpha|0\rangle + \beta|1\rangle$, where $\alpha, \beta \in \mathbb{C}$ and $|\alpha|^2 + |\beta|^2 = 1$. An $n$-qubit state belongs to the tensor product space $(\mathbb{C}^2)^{\otimes n} = \mathbb{C}^{2^n}$, with basis states $|i\rangle_{i \in [2^n]_0}$. When measuring an $n$-qubit state $|\psi\rangle$ in this basis, the outcome $i$ is observed with probability $|\langle i|\psi\rangle|^2$. Quantum operations are described by unitary operators $U$, which satisfy $UU^\dagger = U^\dagger U = I$, where $U^\dagger$ denotes the conjugate transpose and $I$ is the identity operator.

In many quantum algorithms, information is stored and accessed through quantum-read classical-write random access memory (QRAM) (Giovannetti et al., 2008), which is employed in numerous quantum algorithms. QRAM allows the storage and modification of a classical array $a_1, \ldots, a_n$ while supporting quantum query access via the unitary $U_{\text{QRAM}} : |i\rangle|0\rangle \mapsto |i\rangle|a_i\rangle$. If one is only concerned with quantum query complexity, the QRAM assumption can be removed at the cost of a polynomial increase in time complexity. While QRAM serves as a natural quantum analogue of the classical RAM model and is frequently employed in theoretical work, it is important to note that the practical realization of scalable QRAM remains highly uncertain given the current state of quantum hardware.

A quantum (query) algorithm $\mathcal{A}$ is a quantum circuit consisting of a sequence of unitaries $U_1, \ldots, U_T$, where each $U_t$ may represent a quantum gate, a quantum oracle, or a QRAM operation. The time complexity of $\mathcal{A}$ is measured by the total number $T$ of such operations. The algorithm acts on $n$ qubits, initialized in the state $|0\rangle^{\otimes n}$. Applying the unitaries sequentially produces the final state $|\psi\rangle = U_T \ldots U_1 |0\rangle^{\otimes n}$. Finally, a measurement in the computational basis $|i\rangle_{i \in [2^n]_0}$ then yields a classical outcome $i$ with probability $|\langle i|\psi\rangle|^2$.

## B PROOF OF THEOREM 8

### B.1 QUANTUM SUBROUTINES

To present our algorithm more clearly, we introduce the following data structure, which can be viewed as a straightforward application of Theorem 4.

**Proposition 15** (Quantum Modified Leverage Score Approximation). *Assume query access $\mathcal{O}_A$ to a matrix $A = \mathbb{R}^{m \times n}$ with row sparsity $r$ and query access $\mathcal{O}_w$ to a vector $w \in \mathbb{R}_+^m$. For any $0 < \varepsilon \leq 1$, there exists a quantum data structure ModLevApprox that supports the the following operations:*

1. *Initialization: ModLevApprox($\mathcal{O}_A, \mathcal{O}_w, \varepsilon$), outputs an instance $\mathcal{L}$, making $\widetilde{O}(r\sqrt{mn}/\varepsilon)$ queries to $\mathcal{O}_A$ and $\widetilde{O}(\sqrt{mn}/\varepsilon)$ queries to $\mathcal{O}_w$, in $\widetilde{O}(r\sqrt{mn}/\varepsilon + (n^\omega + nr^2)/\varepsilon^2 + n^2/\varepsilon^4)$ time.*

2. *Query: $\mathcal{L}$.Query, outputs an unitary satisfying*

$$\mathcal{L}.\text{Query}|i\rangle|0\rangle = |i\rangle|\widetilde{\sigma}_i\rangle,$$

   *where for all $i \in [m]$, it holds that $(1 - \varepsilon)\sigma_i \leq \widetilde{\sigma}_i \leq (1 + \varepsilon)\sigma_i$, with $\sigma_i = \sigma_i(W^{1/2}A)$ denoting the leverage score, and $W = \text{diag}(w)$. Each query requires $O(r)$ queries to $\mathcal{O}_A$ and $O(1)$ queries to $\mathcal{O}_w$, and runs in $\widetilde{O}(r/\varepsilon^2)$ time.*

*Proof of Proposition 15.* Write $B := W^{1/2}A \in \mathbb{R}^{m \times n}$, where $W = \text{diag}(w)$. Because $W^{1/2}$ rescales each row of $A$ by $\sqrt{w_i}$, $B$ has the *same* row sparsity $r$. Let $A_i$ be the $i$-th row of $A$. Fix a row index $i$ and let $S_i = \text{supp}(A_{i.})$ with $|S_i| = r_i \leq r$. All entries of $A_i$ can be obtained using $O(r)$ queries to $\mathcal{O}_A$, one query to $\mathcal{O}_w$, and $\widetilde{O}(r)$ arithmetic time. For the superposition, we can perform

$$|i\rangle|0\rangle \xrightarrow{\mathcal{O}_A^{\text{idx}}} \frac{1}{\sqrt{r_i}} \sum_{j \in S_i} |i\rangle|j\rangle|0\rangle$$

$$\xrightarrow{\mathcal{O}_A^{\text{elem}}} \frac{1}{\sqrt{r_i}} \sum_{j \in S_i} |i\rangle|j\rangle|A_{ij}\rangle|0\rangle$$

$$\xrightarrow{\mathcal{O}_w} \frac{1}{\sqrt{r_i}} \sum_{j \in S_i} |i\rangle|j\rangle|A_{ij}\rangle|w_i\rangle|0\rangle$$

$$\xrightarrow{U_{\text{MULT}}} \frac{1}{\sqrt{r_i}} \sum_{j \in S_i} |i\rangle|j\rangle|A_{ij}\rangle|w_i\rangle|\sqrt{w_i}A_{ij}\rangle$$

$$\xrightarrow{\mathcal{O}_w^\dagger, \mathcal{O}_A^{\text{elem}\dagger}, \mathcal{O}_A^{\text{idx}\dagger}} \frac{1}{\sqrt{r_i}} \sum_{j \in S_i} |i\rangle|j\rangle|0\rangle|0\rangle|B_{ij}\rangle.$$

Thus one row query in Theorem 4 corresponds $O(r)$ queries to $\mathcal{O}_A$, $O(1)$ queries to $\mathcal{O}_w$, plus $\widetilde{O}(r)$ arithmetic time. Therefore, invoking the Theorem 4 on $B$ with accuracy $\varepsilon$ gives, with high probability: (i) an initialisation that performs $\widetilde{O}(\sqrt{mn}/\varepsilon)$ row queries to $B$ and runs in $\widetilde{O}(r\sqrt{mn}/\varepsilon + (n^\omega + nr^2)/\varepsilon^2 + n^2/\varepsilon^4)$; substituting the row-query cost yields $\widetilde{O}(r\sqrt{mn}/\varepsilon)$ calls to $\mathcal{O}_A$ and $\widetilde{O}(\sqrt{mn}/\varepsilon)$ calls to $\mathcal{O}_w$; (ii) a query unitary that maps $|i\rangle|0\rangle$ to $|i\rangle|\widetilde{\sigma}_i(B)\rangle$ with $(1 - \varepsilon)\sigma_i(B) \leq \widetilde{\sigma}_i(B) \leq (1 + \varepsilon)\sigma_i(B)$, using one row query to $B$. Translating again, each query costs $O(r)$ calls to $\mathcal{O}_A$, $O(1)$ call to $\mathcal{O}_w$, and $\widetilde{O}(r/\varepsilon^2)$ time. □

The following proposition describes the weight update step in the algorithm QMLSO (Algorithm 1).

**Proposition 16** (Weight Computation for Overestimates). *Let $s > 0$. Assume query access $\mathcal{O}_\mathcal{F}$ to the function family $\mathcal{F} = \{f_1, \ldots, f_m\}$ and query access $\mathcal{O}_w$ to a vector $w \in \mathbb{R}_+^m$. Let $\mathcal{L}$ denote an instance of ModLevApprox. Then, there exists a quantum algorithm WeightCompute($\mathcal{O}_\mathcal{F}, \mathcal{O}_w, \mathcal{L}, s$) such that*

$$\text{WeightCompute}(\mathcal{O}_\mathcal{F}, \mathcal{O}_w, \mathcal{L}, s)|i\rangle|0\rangle = |i\rangle|w_i'\rangle$$

*where*

$$w_i' = \frac{1}{s} \cdot \frac{f_i(\sqrt{\sigma_i/w_i})}{\sigma_i/w_i},$$

*and $\sigma_i$ is the leverage score returned by querying $\mathcal{L}$ at index $i \in [m]$. The algorithm makes $O(1)$ queries to $\mathcal{L}$, $\mathcal{O}_w$, and $\mathcal{O}_\mathcal{F}$, and requires an additional $\widetilde{O}(1)$ time for computation.*

*Proof of Proposition 16.* Consider the following procedure:

$$
\begin{aligned}
|i\rangle|0\rangle &\xmapsto{\mathcal{L}.\mathsf{Query}} |i\rangle|\sigma_i\rangle|0\rangle \\
&\xmapsto{\mathcal{O}_w} |i\rangle|\sigma_i\rangle|w_i\rangle|0\rangle \\
&\xmapsto{U_{\mathrm{DIV}}} |i\rangle|\sigma_i\rangle|w_i\rangle|\sigma_i/w_i\rangle|0\rangle|0\rangle \\
&\xmapsto{U_{\mathrm{SQRT}}} |i\rangle|\sigma_i\rangle|w_i\rangle|\sigma_i/w_i\rangle|\sqrt{\sigma_i/w_i}\,\rangle|0\rangle \\
&\xmapsto{\mathcal{O}_\mathcal{F}} |i\rangle|\sigma_i\rangle|w_i\rangle|\sigma_i/w_i\rangle|\sqrt{\sigma_i/w_i}\,\rangle|f_i(\sqrt{\sigma_i/w_i}\,)\rangle \\
&\xmapsto{U_{\mathrm{DIV}(sr)}} |i\rangle|0\rangle|0\rangle|0\rangle|0\rangle|w_i'\rangle,
\end{aligned}
$$

where $U_{\mathrm{DIV}}$ computes the division $\sigma_i/w_i$, $U_{\mathrm{SQRT}}$ computes square root, and $U_{\mathrm{DIV}(sr)}$ carries out the arithmetic $w_i' := \frac{1}{s} \cdot \frac{f_i(\sqrt{\sigma_i/w_i}\,)}{r}$ and then uncomputes the ancillary registers $\sigma_i, w_i, \sigma_i/w_i, \sqrt{\sigma_i/w_i}$.

The computation uses $O(1)$ queries to $\mathcal{L}$, $\mathcal{O}_w$, and $\mathcal{O}_\mathcal{F}$, and $\widetilde{O}(1)$ additional arithmetic operations, as claimed in the proposition. □

The following proposition specifies the output data structure produced by QMLSO (Algorithm 1).

**Proposition 17** (Preparation for Multiscale Overestimates). *Let $0 < \varepsilon < 1$, $\mathcal{J} = \{j_{\min}, \ldots, j_{\max}\} \subseteq \mathbb{Z}$ be a contiguous interval. Suppose we are given a collection of instances $\{\mathcal{L}^{(j)} : j \in \mathcal{J}\}$, where each $\mathcal{L}^{(j)}$ is an instance of $\mathsf{ModLevApprox}$. Then, there exists a quantum data structure $\mathsf{QOverestimate}$ that supports the following operations:*

- *Initialization: $\mathsf{QOverestimate}(\{\mathcal{L}^{(j)} : j \in \mathcal{J}\}, \varepsilon)$, outputs an instance $\mathcal{Z}$ in total time $\widetilde{O}(\sum_{j \in \mathcal{J}} \zeta_j)$, where $\zeta_j$ denotes the needed time of initialization of $\mathcal{L}^{(j)}$.*

- *Query: $\mathcal{Z}.\mathsf{Query}$, outputs an unitary such that for every $i \in [m]$,*

  $$\mathcal{Z}.\mathsf{Query}|i\rangle|0\rangle = |i\rangle|z_i\rangle,$$

  *where*

  $$z_i = \frac{1}{1-\varepsilon} \sum_{j \in \mathcal{J}} \sigma_i^{(j)},$$

  *and $\sigma_i^{(j)}$ is leverage score returned by querying $\mathcal{L}^{(j)}$ at index $i \in [m]$. This query is executed in $\widetilde{O}(\sum_{j \in \mathcal{J}} \iota_j)$ time, where $\iota_j$ represents the time required to querying $\mathcal{L}^{(j)}$.*

*Proof of Proposition 17.* For every $j \in \mathcal{J}$, suppose we hold an instance $\mathcal{L}^{(j)}$ of $\mathsf{ModLevApprox}$. Denote by $\zeta_j$ its initialization cost and by $\iota_j$ the cost of a single query. The initialization of $\mathsf{QOverestimate}$ consists of initializing all instances $\mathcal{L}^{(j)}$ for $j \in \mathcal{J}$ and storing $\varepsilon$ in QRAM in $\widetilde{O}(1)$ time, allowing access through a unitary $U_\varepsilon : |i\rangle|0\rangle \to |i\rangle|1/1-\varepsilon\rangle$.

For the query operation, on input $|i\rangle|0\rangle$, attach an accumulator register initialized to $|0\rangle$, and iterate over the scales $j \in \mathcal{J}$:

$$|i\rangle|0\rangle \xmapsto{\mathcal{L}^{(j_{\min})}.\mathsf{Query}} |i\rangle|\sigma_i^{(j_{\min})}\rangle|0\rangle \xmapsto{U_{\text{ADD}}} |i\rangle|\sigma_i^{(j_{\min})}\rangle|\sigma_i^{(j_{\min})}\rangle|0\rangle$$

$$\xmapsto{(\mathcal{L}^{(j_{\min})}.\mathsf{Query})^{\dagger}} |i\rangle|0\rangle|\sigma_i^{(j_{\min})}\rangle|0\rangle$$

$$\xmapsto{\mathcal{L}^{(j_{\min}+1)}.\mathsf{Query}} |i\rangle|\sigma_i^{(j_{\min}+1)}\rangle|\sigma^{(j_{\min})}\rangle|0\rangle \xmapsto{U_{\text{ADD}}} |i\rangle|\sigma_i^{(j_{\min}+1)}\rangle|\sigma^{(j_{\min})} + \sigma_i^{(j_{\min}+1)}\rangle|0\rangle$$

$$\xmapsto{(\mathcal{L}^{(j_{\min}+1)}.\mathsf{Query})^{\dagger}} |i\rangle|0\rangle|\sigma_i^{(j_{\min})} + \sigma_i^{(j_{\min}+1)}\rangle|0\rangle$$

$$\xmapsto{\cdots} |i\rangle|0\rangle|\sum_{j\in\mathcal{J}} \sigma_i^{(j)}\rangle|0\rangle$$

$$\xmapsto{U_\varepsilon} |i\rangle|1/(1-\varepsilon)\rangle|\sum_{j\in\mathcal{J}} \sigma_i^{(j)}\rangle|0\rangle \xmapsto{U_{\text{MULT}}} |i\rangle|1/(1-\varepsilon)\rangle|\sum_{j\in\mathcal{J}} \sigma_i^{(j)}\rangle|z_i\rangle,$$

The adder $U_{\text{ADD}}$ and multiplier $U_{\text{MULT}}$ are implemented using fixed-point arithmetic and costs $\widetilde{O}(1)$ gates. The query procedure calls each $\mathcal{L}^{(j)}$ once and invokes its inverse once; together this costs $\Theta(\iota_j)$. Summing over all $j \in \mathcal{J}$, the total runtime is $O\left(\sum_{j\in\mathcal{J}} \iota_j\right)$. The arithmetic overhead from the adders contributes $\widetilde{O}(|\mathcal{J}|)$ additional gates, which is absorbed in the stated complexity. $\square$

### B.2 PROOF OF THEOREM 8

We first analyze the time complexity. Since $\theta \in (0,4)$, we have $\delta = \max\left\{\frac{1}{2}, \left|\frac{\theta-2}{2}\right|\right\} < 1$, and thus $T = \Theta_{L,\theta,c}(\log\log\beta)$. In each iteration $i \leq T$, the subroutine ModLevApprox runs in $\widetilde{O}(r\sqrt{mn} + n^\omega + nr^2)$ time (absorbing the cost of queries), and provides query access such that each query requires $O(r)$ calls to $\mathcal{O}_A$, $O(1)$ calls to $\mathcal{O}_{w^{(i)}_{\text{itr}}}$, and $\widetilde{O}(r)$ time (Proposition 15). The subroutine WeightCompute then constructs $U_{w^{(i+1)}_{\text{itr}}}$, where each query requires $\widetilde{O}(r)$ time. This is because it makes $O(1)$ calls to $\mathcal{L}^{(i)}_{\text{itr}}$ and $\mathcal{O}_{w^{(i)}_{\text{itr}}}$, each of which can be executed in $\widetilde{O}(r)$ time (Proposition 16). For the downward recursion across $|\mathcal{J}|$ scales (lines 9-12 of Algorithm 1), the complexity is the same per step. Therefore, the total cost of iterations and recursion is $\widetilde{O}\left((r\sqrt{mn} + n^\omega + nr^2)\Delta\right)$, making $O(\sqrt{mn})$ queries to $\mathcal{O}_{w^\circ}$, $O(r\sqrt{mn}\Delta)$ queries to $\mathcal{O}_A$, $O(\sqrt{mn}\Delta)$ queries to $\mathcal{O}_\mathcal{F}$, where $\Delta = |j_{\max} - j_{\min}| + \max\{\log\log\beta, 0\}$. In the final step of the algorithm, the data structure $\mathcal{Z}$ can be initialized in $\widetilde{O}\left((r\sqrt{mn} + n^\omega + nr^2)|\mathcal{J}|\right)$ time, and each call to $\mathcal{Z}.\mathsf{Query}$ can be executed in $\widetilde{O}(r|\mathcal{J}|)$ time. This follows directly from Proposition 17, with parameters $\zeta_t = \widetilde{O}(r\sqrt{mn} + n^\omega + nr^2)$ and $\iota_t = \widetilde{O}(r)$.

In the remainder of this section, we will show that the vector $z \in \mathbb{R}^m$ corresponding to the output of QMLSO (Algorithm 1) is indeed an $O(n \cdot |j_{\max} - j_{\min}|)$-overestimate with respect to $A$ and an $\alpha$-approximate scheme $\mathcal{W}$. To analyze the correctness of the algorithm, we first introduce the following metric.

**Definition 18.** *Define a metric $d$ on $\mathbb{R}^m_+$ by*

$$d(u, w) := \max\left\{\left|\log\frac{u_i}{w_i}\right| : i \in [m]\right\}$$

It is straightforward to verify that $d$ is indeed a metric, as it satisfies symmetry, positivity, and the triangle inequality.

A useful characterization is the following, which is immediate from the definition: it shows that being an $\alpha$-approximate weight is equivalent to requiring that $w$ is almost a fixed point of the update map $\phi_s$, up to distance $\log\alpha$ in the metric $d$.

**Fact 19.** *Fix a matrix $A \in \mathbb{R}^{m\times n}$ (where the $i$-th row is $a_i$ for each $i \in [m]$) and a family of loss functions $\mathcal{F} = \{f_1, \ldots, f_m\}$. A vector $w \in \mathbb{R}^m_+$ is an $\alpha$-approximate weight (with respect to $A$ and $\mathcal{F}$) at scale $s$ if and only if*

$$d(w, \phi_s(w)) \leq \log\alpha$$

where $\phi_s : \mathbb{R}_+^m \to \mathbb{R}_+^m$ is the update function defined by

$$(\phi_s(w))_i := \frac{1}{s} \cdot \frac{f_i(\sqrt{\sigma_i(W^{1/2}A)/w_i})}{\sigma_i(W^{1/2}A)/w_i}, \quad \text{and} \quad W := \text{diag}(w).$$

Furthermore, $\mathcal{W} = \{w^{(j)} \in \mathbb{R}_+^m : j \in \mathcal{J}\}$ is an $\alpha$-approximate weight scheme if and only if

$$\forall j \in \mathcal{J} : \ d(w^{(j)}, \phi_{2^j}(w^{(j)})) \leq \log \alpha; \tag{6}$$

$$\forall j, j+1 \in \mathcal{J} : \ d(w^{(j+1)}, w^{(j)}) \leq \log \alpha. \tag{7}$$

In the noisy setting (where Algorithm 1 actually operates), we define the update function $\widetilde{\phi}_{s,\varepsilon} : \mathbb{R}_+^m \to \mathbb{R}_+^m$ by

$$\left(\widetilde{\phi}_{s,\varepsilon}(w)\right)_i := \frac{1}{s} \frac{f_i\left(\sqrt{\widetilde{\sigma}_i/w_i}\right)}{\widetilde{\sigma}_i/w_i}, \quad \forall i \in [m], \tag{8}$$

where $\widetilde{\sigma}_i$ is an $\varepsilon$-approximation of $\sigma_i(W^{1/2}A)$, i.e., $(1-\varepsilon)\sigma_i(W^{1/2}A) \leq \widetilde{\sigma}_i \leq (1+\varepsilon)\sigma_i(W^{1/2}A)$.

Our analysis builds on the ideas of Jambulapati et al. (2024). We use the following result, which shows that the weight update procedure remains "contractive" even when noise is present.

**Lemma 20** (Contraction of a noisy update, (Jambulapati et al., 2024, Lemma 2.9)). *Let* $\delta = \max\left\{\frac{1}{2}, \left|\frac{\theta-2}{2}\right|\right\}$, $C = \max\left\{\frac{2L}{c}, \frac{1}{c}\right\}$. *For any* $w, w' \in \mathbb{R}_+^m$ *and* $\varepsilon, \varepsilon' \in [0,1)$, *it holds that* $d(\widetilde{\phi}_{s,\varepsilon}(w), \widetilde{\phi}_{s,\varepsilon'}(w')) \leq \delta \cdot d(w, w') + \log C + \log \frac{1+\varepsilon}{1-\varepsilon'}$.

At the beginning, assuming the initial weight is $\beta$-approximate, we apply the triangle inequality of $d$ together with Lemma 20 to get

$$d\left(w_{\text{itr}}^{(0)}, \widetilde{\phi}_{s_{\max},\varepsilon}\left(w_{\text{itr}}^{(0)}\right)\right) \leq d\left(w_{\text{itr}}^{(0)}, \phi_{s_{\max}}\left(w_{\text{itr}}^{(0)}\right)\right) + d\left(\phi_{s_{\max}}\left(w_{\text{itr}}^{(0)}\right), \widetilde{\phi}_{s_{\max},\varepsilon}\left(w_{\text{itr}}^{(0)}\right)\right)$$

$$\leq \log \beta + \log C + \log(1+\varepsilon).$$

For the iterative procedure (lines 4-7 of Algorithm 1), by applying Lemma 20 inductively, we obtain

$$d\left(w_{\text{itr}}^{(T)}, \widetilde{\phi}_{s_{\max},\varepsilon}\left(w_{\text{itr}}^{(T)}\right)\right) \leq \delta^T \cdot d\left(w_{\text{itr}}^{(0)}, \widetilde{\phi}_{s_{\max},\varepsilon}\left(w_{\text{itr}}^{(0)}\right)\right) + \sum_{i=0}^{T-1} \delta^i \left(\log C + \log \frac{1+\varepsilon}{1-\varepsilon}\right)$$

$$\leq \delta^T \cdot \log \beta + \sum_{i=0}^{T} \delta^i \left(\log C + \log \frac{1+\varepsilon}{1-\varepsilon}\right)$$

$$\leq \frac{\delta^T \log \beta + \log C + \log \frac{1+\varepsilon}{1-\varepsilon}}{1 - \delta}.$$

Therefore, after $T$ iterations, we have

$$d\left(w_{\text{itr}}^{(T)}, \phi_{s_{\max}}\left(w_{\text{itr}}^{(T)}\right)\right) \leq d\left(w_{\text{itr}}^{(T)}, \widetilde{\phi}_{s_{\max},\varepsilon}\left(w_{\text{itr}}^{(T)}\right)\right) + d\left(\widetilde{\phi}_{s_{\max},\varepsilon}\left(w_{\text{itr}}^{(T)}\right), \widetilde{\phi}_{s_{\max},0}\left(w_{\text{itr}}^{(T)}\right)\right)$$

$$\leq d\left(w_{\text{itr}}^{(T)}, \widetilde{\phi}_{s_{\max},\varepsilon}\left(w_{\text{itr}}^{(T)}\right)\right) + \log C + \log(1+\varepsilon)$$

$$\leq \frac{\delta^T \log \beta + \log C + \log \frac{1+\varepsilon}{1-\varepsilon}}{1-\delta} + \log C + \log(1+\varepsilon)$$

$$\leq \frac{\delta^T \log \beta + 2\log C + 2\log(1+\varepsilon) - \log(1-\varepsilon)}{1-\delta}$$

$$\leq \frac{3\log C}{1-\delta},$$

where the first inequality follows from the triangle inequality of $d$, the second inequality applies Lemma 20, the third inequality substitutes the previous bound, and the last inequality follows from the choice of $T$ (line 2 of Algorithm 1).

Set $\alpha := C^{3/(1-\delta)} = \Theta_{L,\theta,c}(1)$. By Fact 19, we have that $w^{(j_{\max})} = w_{\mathrm{itr}}^{(T)}$ is $\alpha$-approximate at scale $s_{\max} = 2^{j_{\max}}$.

We now proceed with a downward recursion across scales (lines 9-12 of Algorithm 1). For each $j = j_{\max} - 1, \ldots, j_{\min}$, define $w^{(j)} := \widetilde{\phi}_{2^j,\varepsilon}(w^{(j+1)})$. By applying Lemma 20 inductively, we obtain that

$$
\begin{aligned}
&d\left(w^{(j)}, \widetilde{\phi}_{2^j,\varepsilon}\left(w^{(j)}\right)\right) \\
&= d\left(\widetilde{\phi}_{2^j,\varepsilon}\left(w^{(j+1)}\right), \widetilde{\phi}_{2^j,\varepsilon}^2\left(w^{(j+1)}\right)\right) \\
&\leq \delta \cdot d\left(w^{(j+1)}, \widetilde{\phi}_{2^j,\varepsilon}\left(w^{(j+1)}\right)\right) + \log C + \log\frac{1+\varepsilon}{1-\varepsilon} \\
&= \delta \cdot \left(d\left(w^{(j+1)}, \widetilde{\phi}_{2^{j+1},\varepsilon}\left(w^{(j+1)}\right)\right) + \log 2\right) + \log C + \log\frac{1+\varepsilon}{1-\varepsilon} \\
&\leq \delta^{j_{\max}-j} \cdot \left(d\left(w^{(j_{\max})}, \widetilde{\phi}_{s_{\max},\varepsilon}\left(w^{(j_{\max})}\right)\right) + \log 2\right) + \sum_{i=0}^{j_{\max}-1-j} \delta^i\left(\log C + \log\frac{1+\varepsilon}{1-\varepsilon}\right) \\
&\leq \log\alpha + \log 2 + \frac{\log C + \log\frac{1+\varepsilon}{1-\varepsilon}}{1-\delta}.
\end{aligned}
$$

Using the above result, we can now verify that Eq. (6) holds.

$$
\begin{aligned}
d\left(w^{(j)}, \phi_{2^j}\left(w^{(j)}\right)\right) &= d\left(w^{(j)}, \widetilde{\phi}_{2^j,\varepsilon}\left(w^{(j)}\right)\right) + d\left(\widetilde{\phi}_{2^j,\varepsilon}\left(w^{(j)}\right), \widetilde{\phi}_{2^j,0}\left(w^{(j)}\right)\right) \\
&\leq d\left(w^{(j)}, \widetilde{\phi}_{2^j,\varepsilon}\left(w^{(j)}\right)\right) + \log C + \log(1+\varepsilon) \\
&\leq \log\alpha + \log 2 + \frac{\log C + \log\frac{1+\varepsilon}{1-\varepsilon}}{1-\delta} + \log C + \log(1+\varepsilon) \\
&\leq \log\alpha + \log 2 + \frac{2\log C + 2\log(1+\varepsilon) - \log(1-\varepsilon)}{1-\delta} \\
&\leq 2\log\alpha + \log 2.
\end{aligned}
$$

And confirm that Eq. (7) is satisfied.

$$
d\left(w^{(j+1)}, w^{(j)}\right) = d\left(w^{(j+1)}, \widetilde{\phi}_{2^j,\varepsilon}\left(w^{(j+1)}\right)\right) = d\left(w^{(j+1)}, \widetilde{\phi}_{2^{j+1},\varepsilon}\left(w^{(j+1)}\right)\right) + \log 2 \leq 2\log 2\alpha.
$$

Therefore, $\mathcal{W} := \left\{w^{(j)} : j \in \mathcal{J} := \mathbb{Z} \cap [j_{\min}, j_{\max}]\right\}$ forms a $4\alpha^2$-approximate weight scheme. It remains to verify that the vector $z$ in line 12 of Algorithm 1 (see Proposition 17 for details) is an $O(n \cdot |j_{\max} - j_{\min}|)$-overestimate with respect to $\mathcal{W}$.

For every $i \in [m]$, note that

$$
z_i = \frac{1}{1-\varepsilon}\sum_{j\in\mathcal{J}}\widetilde{\sigma}_i\left(W_j^{1/2}A\right) \geq \sum_{j\in\mathcal{J}}\sigma_i\left(W_j^{1/2}A\right) \geq \max_{j\in\mathcal{J}}\sigma_i(W_j^{1/2}A),
$$

and

$$
\begin{aligned}
\|z\|_1 = \sum_{i\in[m]}z_i &= \frac{1}{1-\varepsilon}\sum_{i\in[m]}\sum_{j\in\mathcal{J}}\widetilde{\sigma}_i\left(W_j^{1/2}A\right) \leq \frac{1+\varepsilon}{1-\varepsilon}\sum_{j\in\mathcal{J}}\sum_{i\in[m]}\sigma_i\left(W_j^{1/2}A\right) \\
&\leq \frac{1+\varepsilon}{1-\varepsilon}\sum_{j\in\mathcal{J}}\mathrm{Tr}(W_j^{1/2}A\left(A^\top W_j A\right)^+\left(W_j^{1/2}A\right)^\top) \leq \frac{1+\varepsilon}{1-\varepsilon}\sum_{j\in\mathcal{J}}\mathrm{rank}(A) \\
&\leq \frac{1+\varepsilon}{1-\varepsilon}(j_{\max} - j_{\min})n.
\end{aligned}
$$

## C  PROOF OF THEOREM 9

*Proof of Theorem 9.* The construction follows the same threestep recipe as in the classical argument of Jambulapati et al. (2024): (1) pick anchor points $\hat{t}_i$ on each loss function; (2) use ModLevApprox

to estimate leverage scores at those anchors and convert them into weights; (3) add a quadratic bump to every $f_i$ to obtain a better-behaved family $\mathcal{F}^\circ$ together with an approximate weight vector $w^\circ$.

**Step 1: Anchor Selection.** Since each $f_i$ is $(L, \theta, c)$-proper, we can efficiently search over $t > 0$ to locate anchor points $\widehat{t}_i$ satisfying

$$\frac{1}{2} s_{\max} \le f_i(\widehat{t}_i) \le s_{\max}, \quad i \in [m],$$

as detailed in Proposition 21. Thus, a quantum oracle $\mathcal{O}_t$ to these anchors can be constructed using $O(1)$ queries to $\mathcal{O}_\mathcal{F}$.

**Step 2: Quantum Leverage Score Approximation.** Invoke $\mathsf{ModLevApprox}(\mathcal{O}_A, \mathcal{O}_h, \frac{1}{2})$, where $\mathcal{O}_h$ queries the vector $h = (\widehat{t}_1^{-2}, \ldots, \widehat{t}_m^{-2})$, constructed in $\widetilde{O}(1)$ time via $O(1)$ queries to $\mathcal{O}_t$. The quantum algorithm uses $\widetilde{O}(r\sqrt{mn})$ queries to $\mathcal{O}_A$, $\widetilde{O}(\sqrt{mn})$ queries to $\mathcal{O}_\mathcal{F}$, and runs in $\widetilde{O}(r\sqrt{mn} + n^\omega + nr^2)$ time. It outputs an approximation $\widetilde{\sigma}$ satisfying $\frac{1}{2}\sigma(H^{1/2}A) \le \widetilde{\sigma}_i \le \frac{3}{2}\sigma(H^{1/2}A)$ for $H = \mathrm{diag}(h)$. Set

$$w_i^\circ := \delta/\widetilde{\sigma}_i, \quad i \in [m],$$

and define query access $\mathcal{O}_{w^\circ}$, where each query requires $O(r)$ queries to $\mathcal{O}_A$, $O(1)$ queries to $\mathcal{O}_\mathcal{F}$, and takes $\widetilde{O}(r)$ time.

**Step 3: Modified Loss Definition.** Define modified losses as:

$$f_i^\circ(t) = f_i(t) + s_{\max} w_i^\circ \cdot t^2, \quad \forall i \in [m].$$

Following Jambulapati et al. (2024, Fact 2.10), each $(f_i^\circ)^{1/2}$ is $\max\{1, L\}$-auto-Lipschitz, $\theta$-lower-homogeneous with constant $c$, making $\mathcal{F}^\circ$ a $(\max\{1, L\}, \theta, c)$-proper family. The correctness of Eq. (4) and that $w^\circ$ is $\beta$-approximate at scale $s_{\max}$ directly follows from Jambulapati et al. (2024, Theorem 2.11), yielding $C_{\mathrm{INIT}} = 2(2L/c)^{2/\theta}$. □

**Proposition 21.** *Let $\mathcal{O}_f$ be an oracle for a $(L, \theta, c)$-proper function $f : \mathbb{R} \to \mathbb{R}_+$. For any $s_{\max} > s_{\min} > 0$, there exists an algorithm $\mathsf{FindSol}(\mathcal{O}_f, s_{\min}, s_{\max})$ that outputs $x > 0$ such that $s_{\min} \le f(x) \le s_{\max}$, making at most $O_{L,\theta,c}\left(\log \frac{s_{\max}}{s_{\min}}\right)$ queries to $\mathcal{O}_f$, and running in $O_{L,\theta,c}\left(\log \frac{s_{\max}}{s_{\min}}\right)$ time.*

*Proof of Proposition 21.* Let $h = \sqrt{f}$. Recall Definition 2, $h$ is $L$-auto-Lipschitz and lower $\theta$-homogeneous with constant $c > 0$. We implement oracle access to $h$ via a single call to $\mathcal{O}_f$ and a square root, so the query complexity to $\mathcal{O}_h$ equals that to $\mathcal{O}_f$. Let $h_{\min} := \sqrt{s_{\min}}$, $h_{\max} := \sqrt{s_{\max}}$. Our goal is to find $x > 0$ such that $h_{\min} \le h(x) \le h_{\max}$.

We first query $h(1)$. If $h_{\min} \le h(1) \le h_{\max}$, we are done. Otherwise, we use exponential search to find an interval $[a, b]$ (with $b = 2a$ or $a = 2b$) such that $h$ crosses the range $[h_{\min}, h_{\max}]$ within this bracket.

**Case 1:** $h(1) < h_{\min}$. By lower $\theta$-homogeneity,

$$h(2^j) \ge c \cdot 2^{j\theta} h(1), \qquad j \ge 1.$$

Thus, for $j^\star = \left\lceil \frac{1}{\theta} \log_2 \frac{h_{\max}}{c \cdot h(1)} \right\rceil$, we have $h(2^{j^\star}) \ge h_{\max}$. Therefore, within $O(\frac{1}{\theta} \log \frac{h_{\max}}{c \cdot h(1)})$ queries we either find $x$ with $h(x) \in [h_{\min}, h_{\max}]$ or obtain a bracket $[2^{j^\star - 1}, 2^{j^\star}]$.

**Case 2:** $h(1) > h_{\max}$. For $j \ge 1$, taking $\lambda = 2^j$ and $x = 2^{-j}$ in the homogeneity property gives

$$h(1) = h(2^j \cdot 2^{-j}) \ge c \cdot 2^{j\theta} h(2^{-j}) \quad \Rightarrow \quad h(2^{-j}) \le \frac{h(1)}{c \cdot 2^{j\theta}}.$$

Hence for $j^\star = \left\lceil \frac{1}{\theta} \log_2 \frac{h(1)}{c \cdot h_{\min}} \right\rceil$, we have $h(2^{-j^\star}) \le h_{\min}$, and again we either hit the target range or obtain a bracket $[2^{-j^\star}, 2^{-(j^\star - 1)}]$.

**Bisection step.** Assume w.l.o.g. we have a bracket $[a, b]$ with $b = 2a$ and $h(a) \leq h_{\min} < h_{\max} \leq h(b)$ (the mirrored case is analogous). Let $\Delta := h_{\max} - h_{\min} > 0$ and let the half-length be $\delta_0 := \frac{b-a}{2} = \frac{a}{2}$. For $t \geq 0$, let the half-length after $t$ bisections be $\delta_t := \delta_0/2^t$. By lower $\theta$-homogeneity (applied with $x = \delta_{t+1}$ and $\lambda = 2$),

$$h(\delta_{t+1}) \leq \frac{1}{c2^\theta} h(\delta_t) \quad \Rightarrow \quad h(\delta_t) \leq \left(\frac{1}{c2^\theta}\right)^t h(\delta_0).$$

Moreover, since $b - a = a$ and $h(a) \leq h_{\min}$, we have

$$h(\delta_0) = h\left(\frac{a}{2}\right) \leq \frac{1}{c2^\theta} h(a) \leq \frac{h_{\min}}{c2^\theta}.$$

Combining the two displays yields

$$Lh(\delta_t) \leq L\left(\frac{1}{c2^\theta}\right)^{t+1} h_{\min}.$$

Choose

$$t \geq \left\lceil \frac{1}{\theta + \log_2 c} \log_2\left(\frac{Lh_{\min}}{\Delta}\right) \right\rceil - 1.$$

Then $Lh(\delta_t) \leq \Delta$. Let $m$ be the midpoint of the current bracket $[a', b']$ (of length $\delta_{t-1}$). By the $L$-auto-Lipschitz property,

$$h(m) \leq h(a') + Lh(\delta_t) \leq h_{\min} + \Delta = h_{\max},$$
$$h(m) \geq h(b') - Lh(\delta_t) \geq h_{\max} - \Delta = h_{\min},$$

so $h(m) \in [h_{\min}, h_{\max}]$ and the midpoint succeeds.

Overall, the total query complexity of FindSol is $O_{L,\theta,c}\left(\log \frac{s_{\max}}{s_{\min}}\right)$, under the assumption that $f(1)$ is bounded by $\mathrm{poly}(s_{\min}, s_{\max})$. The time is linear in the number of queries.

□

## D   PROOF OF THEOREM 10

Before proving Theorem 10, we restate two quantum algorithms that are invoked as subroutines in our algorithm.

**Theorem 22** (Multiple Quantum State Preparation, Theorem 1 in Hamoudi (2022)). *Let $1 \leq k \leq n$. There exists a quantum algorithm MultiSample($\mathcal{O}_w, k$) that, given query access $\mathcal{O}_w$ to a nonnegative vector $w \in \mathbb{R}_+^n$, outputs with high probability a sample sequence $\vartheta \in [n]^k$ in which each element $i$ is drawn with probability proportional to $w_i$. The algorithm makes $\widetilde{O}(\sqrt{kn})$ queries to $\mathcal{O}_w$ and runs in $\widetilde{O}(\sqrt{kn})$ time.*

**Theorem 23** (Quantum Sum Estimation, (Li et al., 2019, Lemma 3.1)). *There exists a quantum algorithm SumEstimate($\mathcal{O}_w, \varepsilon$) that, given query access $\mathcal{O}_w$ to a nonnegative vector $w \in \mathbb{R}_+^n$, outputs with high probability an $\varepsilon$-approximation $\widetilde{s}$ to the sum $s = \sum_{i \in [n]} w_i$. The algorithm uses $\widetilde{O}(\sqrt{n}/\varepsilon)$ queries to $\mathcal{O}_w$ and runs in $\widetilde{O}(\sqrt{n}/\varepsilon)$ time.*

We will use the following result from Jambulapati et al. (2024) in a black-box manner.

**Theorem 24.** *Let $F(x) := \sum_{i=1}^m f_i(\langle a_i, x \rangle)$ be a total loss function defined by vectors $a_1, \ldots, a_m \in \mathbb{R}^n$ and a $(L, \theta, c)$-proper loss functions family $\mathcal{F} = \{f_1, \ldots, f_m\}$. Let $0 \leq s_{\min} < s_{\max}$. Given any $O_{L,\theta,c}(1)$-approximate weight scheme $\mathcal{W}$ with respect to matrix $A$ (with rows $a_i$), loss family $\mathcal{F}$, and interval $\mathcal{J} := [\log_2 s_{\min} - 4\log m, \log_2 s_{\max}] \bigcap \mathbb{Z}$, with high probability, importance sampling each function $f_i \in \mathcal{F}$ with probability*

$$p_i \geq \max_{j \in \mathcal{J}} \sigma_i(W_j^{1/2} A)$$

*yields an $s$-sparse $\varepsilon$-approximate sparsifier of $F$ over the range $[s_{\min}, s_{\max}]$, where*

$$s = O(\|p\|_1 \log^3 m/\varepsilon^2).$$

*Proof of Theorem 10.* According to Theorem 9, we obtain query access to $\mathcal{F}^\circ$ and $w^\circ$ in $\widetilde{O}(r\sqrt{mn} + n^\omega + nr^2)$ time, and each query costs $\widetilde{O}(r)$ time. These satisfy the property that $w^\circ$ is an $O(m^4 s_{\max}/(\varepsilon s_{\min}))$-approximate weight at scale $s_{\max}$ with respect to $\mathcal{F}^\circ$, and moreover $\mathcal{F}^\circ$ satisfies the condition

$$\sum_{i=1}^m f_i\left(\langle a_i, x\rangle\right) \leq s_{\max} \quad \Rightarrow \quad 0 \leq \sum_{i=1}^m f_i^\circ\left(\langle a_i, x\rangle\right) - f_i\left(\langle a_i, x\rangle\right) \leq \frac{C_{\text{INIT}}\varepsilon s_{\min}}{m}. \tag{9}$$

Note that $|\mathcal{J}| = j_{\max} - j_{\min} = O(\log(ms_{\max}/s_{\min})) = \widetilde{O}(\log(s_{\max}/s_{\min}))$. By applying Theorem 8, we obtain query access to a multiscale $O(n\log(s_{\max}/s_{\min}))$-overestimates $z$ with respect to $\mathcal{F}^\circ$. This construction requires $\widetilde{O}\left((r\sqrt{mn} + n^\omega + nr^2)\log(s_{\max}/s_{\min})\right)$ time, and each query can be executed in $\widetilde{O}(r\log(s_{\max}/s_{\min}))$ time. Combining this with the multiple quantum state preparation (Theorem 22), we can generate a sequence $\vartheta \in [m]^M$ in $\widetilde{O}(\sqrt{Mm} \cdot \log(s_{\max}/s_{\min})) = \widetilde{O}(r\sqrt{mn}\log(s_{\max}/s_{\min})/\varepsilon)$ time. The quantum sum estimation procedure (Theorem 23) then runs in $\widetilde{O}(r\sqrt{m}\log(s_{\max}/s_{\min}))$ time, producing $\widetilde{\nu}$ as a 0.1-approximation of $\nu = \|z\|_1$. Therefore, the overall time complexity is $\widetilde{O}\left((n^\omega + nr^2 + r\sqrt{mn}/\varepsilon) \cdot \log(s_{\max}/s_{\min})\right)$.

It remains to show that

$$\widetilde{F}(x) := \sum_{i=1}^m \frac{\#\{k \in [M] : \vartheta_k = i\}}{1.1 M z_i/\widetilde{\nu}} f_i(\langle a_i, x\rangle)$$

is an $\widetilde{O}(\log(s_{\max}/s_{\min}) \cdot n/\varepsilon^2)$-sparse $\varepsilon$-approximate sparsifier of $F$ over the range $[s_{\min}, s_{\max}]$.

In each sampling round (out of $M$ total rounds), the probability of selecting each $f_i \in \mathcal{F}$ can be expressed (after reweighting) as

$$1.1 z_i \cdot \frac{\nu}{\widetilde{\nu}} \geq \sigma_i^{\text{ms}}(A, \mathcal{W}^\circ) \geq \max_{j \in \mathcal{J}} \sigma(W^{\circ 1/2}_j A),$$

where $W^\circ$ is an $O(1)$-approximate weight scheme with respect to $\mathcal{F}^\circ$. The inequalities follow from the bound $\widetilde{\nu} \leq 1.1\nu$ and from Definition 7.

Applying Theorem 24, we obtain that $\widetilde{F}(x)$ is an $O(\|z\|_1 \log^3 m/\varepsilon) = \widetilde{O}(n\log(s_{\max}/s_{\min})/\varepsilon^2)$-sparse $\varepsilon$-approximate sparsifier of $F^\circ$ over the range $[s_{\min}, s_{\max}]$. Finally, by Eq. (9), for all $x$ such that $s_{\min} \leq F(x) \leq s_{\max}$, we have

$$\left|\widetilde{F}(x) - F(x)\right| \leq \left|\widetilde{F}(x) - F^\circ(x)\right| + |F^\circ(x) - F(x)|$$

$$\overset{Eq.\ (9)}{\leq} \varepsilon F^\circ(x) + \frac{C_{\text{INIT}}\varepsilon s_{\min}}{m}$$

$$\overset{Eq.\ (9)}{\leq} \varepsilon\left(F(x) + \frac{C_{\text{INIT}}\varepsilon s_{\min}}{m}\right) + \frac{C_{\text{INIT}}\varepsilon s_{\min}}{m}$$

$$\leq \varepsilon F(x) + \frac{C_{\text{INIT}}(\varepsilon^2 + \varepsilon)}{m} F(x)$$

$$\leq 2\varepsilon F(x).$$

By setting $\varepsilon \leftarrow \frac{\varepsilon}{2}$, we obtain what we desired. $\qquad\square$

# E  APPLICATIONS

As a special case of $\ell_p$ regression with $p = 2$, the multiscale leverage score overestimates reduce to the leverage score overestimates. In this case, we can directly apply quantum leverage score approximation (Theorem 4) with constant accuracy, without invoking QMLSO (Algorithm 1). Combining the classical linear regression algorithm (Clarkson & Woodruff, 2017, Theorem 43) with our quantum sparsification framework yields the following corollary.

**Corollary 25** (Quantum Linear Regression). *There exists a quantum algorithm that, given query access to a matrix $A \in \mathbb{R}^{m \times n}$ (with row sparsity $r \leq n$) and vector $b \in \mathbb{R}^m$, and $\varepsilon > 0$, outputs with high probability an $x \in \mathbb{R}^n$ such that $\|Ax - b\|_2 \leq (1+\varepsilon)\min_{x \in \mathbb{R}^n}\|Ax - b\|_2$, in $\widetilde{O}(r\sqrt{mn}/\varepsilon + n^3)$ time.*

Note that the sparsifier size is $m' = \widetilde{O}(n/\varepsilon^2)$, which is smaller than the original size $m$, implying $\varepsilon = \Omega(\sqrt{n/m})$. Consequently, the classical runtime term $\widetilde{O}(mr)$ reduces to $\widetilde{O}(m'r) = \widetilde{O}(nr/\varepsilon^2) = \widetilde{O}(r\sqrt{mn}/\varepsilon)$.

As discussed in Section 4.2, multiple regression can be reformulated as linear regression through vectorization. Accordingly, we first apply quantum leverage score approximation (Theorem 4) to the matrix $A \in \mathbb{R}^{m \times n}$ and perform importance sampling to obtain a reduced matrix $A'$, of size $\widetilde{O}(n/\varepsilon^2) \times n$. Finally, we apply the classical multiple regression algorithm (e.g., (Clarkson & Woodruff, 2017, Theorem 38)) to $A'$ to compute the solution.

**Corollary 26** (Quantum Multiple Regression). *There exists a quantum algorithm that, given query access to a matrix $A \in \mathbb{R}^{m \times n}$ (with row sparsity $r \leq n$) and matrix $B \in \mathbb{R}^{m \times N}$, and $\varepsilon > 0$, outputs with high probability an $X \in \mathbb{R}^{n \times N}$ such that $\|AX - B\|_F \leq (1 + \varepsilon) \min_{x \in \mathbb{R}^n} \|AX - B\|_F$, in $\widetilde{O}\left(r\sqrt{mn}/\varepsilon + n^2(Nn + n/\varepsilon + N/\varepsilon)\right)$ time.*

Ridge regression can be reduced to linear regression by augmenting with $n$ additional data points (see Section 4.2).

**Corollary 27** (Quantum Ridge Regression). *Let $\lambda > 0$ be a regularization parameter. There exists a quantum algorithm that, given query access to a matrix $A \in \mathbb{R}^{m \times n}$ (with row sparsity $r \leq n$) and vector $b \in \mathbb{R}^m$, and $\varepsilon > 0$, outputs with high probability an $x \in \mathbb{R}^n$ such that $\|Ax - b\|_2^2 + \lambda \|x\|_2^2 \leq (1 + \varepsilon) \min_{x \in \mathbb{R}^n} \left(\|Ax - b\|_2^2 + \lambda \|x\|_2^2\right)$, in $\widetilde{O}(r\sqrt{mn}/\varepsilon + n^3)$ time.*

As discussed in Section 4.2, lasso regression can also be embedded into the quantum GLM sparsification by introducing $n$ augmented data points and assigning suitable loss functions $f_i$ to them.

**Corollary 28** (Quantum Lasso Regression). *Let $\lambda > 0$ be a regularization parameter. There exists a quantum algorithm that, given query access to a matrix $A \in \mathbb{R}^{m \times n}$ (with row sparsity $r \leq n$) and vector $b \in \mathbb{R}^m$, and $\varepsilon > 0$, outputs with high probability an $x \in \mathbb{R}^n$ such that $\|Ax - b\|_2^2 + \lambda \|x\|_1 \leq (1 + \varepsilon) \min_{x \in \mathbb{R}^n} \left(\|Ax - b\|_2^2 + \|x\|_1\right)$, in $\widetilde{O}(r\sqrt{mn}/\varepsilon + n^3/\varepsilon^2)$ time.*

