# OpenReview forum: "Accelerating Regression Tasks with Quantum Algorithms"
_ICLR.cc/2026/Conference — ICLR 2026 Conference Withdrawn Submission_

### Official Review · Reviewer_aYYA · 2025-10-31

**Soundness:** 2
**Presentation:** 3
**Contribution:** 2
**Rating:** 2
**Confidence:** 1

**Summary:**

This paper introduces a unified quantum framework for accelerating sparsification in generalized linear models (GLMs), yielding quadratic speedups over classical methods across linear, multiple, Lasso, Ridge, Huber, $\ell_p$, and $\gamma_p$ regression. The authors extend the recent classical breakthrough of Jambulapati et al. (2024) by developing a Quantum Multiscale Leverage Score Overestimates (QMLSO) algorithm that leverages quantum leverage score approximation and quantum state preparation techniques. For problem dimension $n$, sparsity $r<n$, sample size $m$, and error parameter $\varepsilon$, their algorithm achieves $\widetilde{\mathcal{O}}!\left(\frac{r\sqrt{mn}}{\varepsilon}+\mathrm{poly}!\left(n,\frac{1}{\varepsilon}\right)\right)$ quantum time complexity, demonstrating quadratic speedup in m when $\varepsilon$ is constant and $m \gg n$.

**Strengths:**

1. Comprehensive theoretical framework with broad applicability. The paper successfully unifies quantum speedups for a diverse family of regression problems under a single GLM sparsification framework, extending beyond previous quantum work that focused primarily on linear regression. The introduction of proper loss functions and multiscale leverage score overestimates provides elegant abstractions that capture the essential structure needed for quantum acceleration across multiple problem variants.
2. Rigorous technical development with novel quantum subroutines. The QMLSO algorithm (Algorithm 1) and the quantum weight initialization procedure (Theorem 9) represent substantial technical contributions that carefully adapt classical contractive algorithms to the quantum setting. The complexity analysis is thorough, properly accounting for query complexity to different oracles and demonstrating clear quadratic speedups with explicit leading-order terms that dominate in the regime where sparsification is beneficial $\varepsilon=\Omega!\left(\sqrt{\tfrac{n}{m}}\right)$.

**Weaknesses:**

1. Limited discussion of QRAM requirements. While the paper acknowledges on page 5 (lines 217-220) that "QRAM serves as a natural quantum analogue of the classical RAM model" and mentions that "practical realization of scalable QRAM remains highly uncertain," there is insufficient critical analysis of how QRAM requirements scale with problem size and what this means for near-term quantum advantage.

2. Gap between stated contributions and actual novelty over prior quantum work. The paper claims on page 1 (lines 17-18) to address "broader and both theoretically and practically important regression tasks such as ℓp regression and Huber regression" beyond linear regression, but the relationship to Song et al. (2023) is understated. As acknowledged in Table 1, Song et al. already achieved $\widetilde{\mathcal{O}}!\left(\frac{\sqrt{m},n^{1.5}}{\varepsilon}\right)$ time for linear, multiple, and ridge regression, where the improvement here is from $n^{1.5}$ to $rn$, which is significant only when $r \ll n^{0.5}$.

**Questions:**

1. Can you provide explicit bounds on the QRAM size, access time, and circuit depth required for representative problem instances?

2. Can you provide a revised complexity analysis that explicitly accounts for QRAM access time as a function of the data size, and identify at what point (if any) QRAM overhead eliminates the quantum advantage?

3. What exactly is the time complexity of your algorithms without QRAM? Does the quantum advantage survive this modification, and if so, under what parameter regimes?

---

> ### Comment · Reviewer_aYYA · 2025-11-19
>
> Since I am not an expert in this area, so I seriously read other reviewers feedback. After carefully considering the other reviewers’ feedback, I would like to increase the score to 4.

---

### Official Review · Reviewer_C66o · 2025-10-31

**Soundness:** 3
**Presentation:** 3
**Contribution:** 2
**Rating:** 4
**Confidence:** 3

**Summary:**

This paper presents a quantum framework for efficiently solving various regression and optimization problems by adapting Jambulapati et. al. into a quantum algorithm and noticing that many of the regression problems can be formulated as sparse generalized linear models. It achieves quadratic speedup comparing to previously best-known algorithms.

**Strengths:**

- This paper provides a unified optimization framework for many of the regression problems in optimization, which provides a convenient way for further quantum algorithm research.
- The paper itself is presented well, with easy-to-follow narratives.

**Weaknesses:**

- The paper does not highlight clearly the source of the quantum speedup, which might be hard to understand. Also the paper, although citing the Jambulapati et. al., does not discuss in detail how the classic algorithm is related to the quantum version.
- The contribution of the paper is a bit lacking. Although a unified speedup across different regression problems is nice, the contribution of the paper is limited to implementing the original algorithm with several quantum tricks (like Hamoudi's copy preparation trick, which is well-known and widely applied, for example in https://arxiv.org/abs/2402.12745)

**Questions:**

- Is it possible to prove a quantum lower bound on the regression problems in the paper?
- Is it possible to incorporate also infinite-norm regression (i.e., max) into the framework?

---

> ### Author Response · Authors · 2025-11-15
>
> We thank the reviewer for the constructive feedback and for the positive comments on our framework and presentation.
>
> Regarding the source of quantum speedup: as stated in the Techniques subsection of the introduction, the quantum speedup in our GLM framework comes from two parts: first, efficiently computing suitable importance scores, and second, performing importance sampling based on these scores. For the sampling step, we use Hamoudi's copy-preparation trick, which is indeed a powerful and widely used technique. We will expand our discussion of this subroutine and add further references, including arXiv:2402.12745. Our main technical contribution lies more in the first part: for much more general loss functions, providing efficient quantum access to the required importance scores is highly nontrivial. This requires carefully studying the most advanced and technically involved classical results and then designing how to use existing quantum tools in a precise and efficient way to implement them. In addition, we deliberately prioritized readability, striving to present the concrete structure of the quantum algorithms and the proof ideas as clearly as possible, in order to lay a transparent foundation for broader applications of these techniques in the quantum algorithms community.
>
> Regarding quantum lower bounds: we briefly record two complementary results that apply to our setting. First, note that linear regression is a special case of GLM sparsification (by taking $f_i = \ell_2$ for all $i$). Hence, any lower bound for quantum spectral approximation immediately implies a corresponding lower bound for quantum GLM sparsification.The first lower bound establishes the optimal $m$-dependence for constant $\varepsilon$: for arbitrary $m \ge n$, there exists a family of matrices $A \in \mathbb{R}^{m\times n}$ such that any quantum algorithm that computes a constant-factor approximation of $F$ must make $\Omega (\sqrt{mn})$ row queries to $A$ (Theorem 8.1,  Apers and Gribling (2023) ). The second lower bound establishes the optimal $\varepsilon$-dependence in the regime $m \in O (n^2)$: for arbitrary $n$ and $m \in O(n^2)$ and any $\varepsilon \in \Omega (\sqrt{m/n})$, there exists a family of matrices $A \in \mathbb{R}^{m\times n}$ such that any quantum algorithm that computes an $\varepsilon$-approximation of $F$ must make $\Omega(\sqrt{mn}/\varepsilon)$ row queries to $A$ (Theorem 2, Apers and de Wolf (2020)  ).
>
> Regarding infinite-norm regression: since the objective there uses a max aggregator rather than a sum, it does not directly fall under our current GLM template, which is built around a sum-aggregated loss. Nevertheless, one could in principle approximate the $\ell_\infty$ objective by leveraging the duality structure of $\ell_p$ losses for $p \in (0,2]$ together with a large-$p$ approximation scheme for $\ell_p$ regression. This would require a careful control of the dependence on $p$ and a delicate approximation analysis, which goes beyond the scope of the present work and which we leave as an interesting direction for future research.
>
>
> We hope that these clarifications help convey the full scope and novelty of our contributions, and we would be grateful if you could take them into account in your overall assessment.

---

> ### Comment · Reviewer_aYYA · 2025-11-20
> **For $\ell_{\infty}$ regression discussed by Reviewer C66o and Authors**
>
> I think the second question asked by Reviewer C66o ''Is it possible to incorporate also infinite-norm regression (i.e., max) into the framework?'' is a super interesting questions. First of all, the $\ell_{\infty}$ regression has two potential definitions at least. One is just setting $p = \infty$ for $\ell_p$ regression formulation $\\| A x- b \\|\_{\infty}$,
>  the other case is you want to find an $x$ such that $\\| x - x^* \\|\_{\infty}$.
>
> For the first case. Usually in theory applications, there are some cases we want the $\ell_{\infty}$-norm, for example, the John Ellipsoid computation [1] (which is using $\ell_{\infty}$-Lewis weights). However, in some linear programming applications, just usually just $p=\log(m/n)$ or $\log m$ is good enough (see page 7, the sentences before Section 1.4 in [2]).
>
> For the second case, previous work [3] show that in ``sketch-and-solve'' regime, in order to get good result, the sketching matrices have to be the dense ones e.g., subsampled randomized hadamard transform. It has been known that leverage score such diagonal sampling matrix (with small number of non-zeros on diagonal) is not going to give a good result, even in classical (see Theorem 3 in [3] provided a lower bound).
>
> [1] Michael B Cohen, Ben Cousins, Yin Tat Lee, and Xin Yang. A near-optimal algorithm for approximating the john ellipsoid. COLT 2019
>
> [2] Yin Tat Lee, Aaron Sidford. Solving Linear Programs with Sqrt(rank) Linear System Solves. arXiv 2019, 1910.08033
>
> [3] Eric Price, Zhao Song, David P. Woodruff, Fast Regression with an $\ell_{\infty}$ Guarantee. ICALP 2017

---

### Official Review · Reviewer_Ewfu · 2025-10-31

**Soundness:** 3
**Presentation:** 1
**Contribution:** 2
**Rating:** 2
**Confidence:** 3

**Summary:**

The paper proposes a quantum algorithm for improving the speed of finding sparsifiers (most commonly, weighted subset of training samples) that approximately preserve the value of the loss function over the parameter domain, allowing for faster training.

**Strengths:**

The paper applies recent work on quantum leverage score approximation by Apers and Gribling in the context of recent result showing existence of sparsifiers for a wide class of losses in Generalized Linear Models\ by Jambulapati et al. The resulting approach improves the complexity of finding the sparsifier: it shares the $\sqrt{m}$ term with existing approaches, but improves the dependence on $n$ from $n^{1.5}$ to $rn^{0.5}$, an advantage in sparse ($r<<n$) scenarios.

**Weaknesses:**

The result is an incremental advancement combining existing classical work (Jambulapati et al.) with prior quantum algorithms (Hamoudi, Li et al., Apers and Gribling). The presentation in the manuscript makes it unclear which results are prior work and which are novel (e.g. Def. 5. and Def. 6 should include reference to Jambulapati et al.). The main manuscript provides no clear description of the algorithms beyond listing the steps in Alg 1. and Alg. 2, and many of the key elements in these two algorithms are not properly explained. For example, in Alg. 1, ModLevApprox and WeightCompute are jointly described using one, very high-level sentence in the main manuscript; description in the appendix B.1. indicates these are not original contributions, but recapitulation of Apers and Gribling’s Theorem 4.

The algorithm is aimed at a very practical problem of speeding up GLM, however, by working in a QRAM-based framework, it’s practical applicability on near-to-medium term hardware is heavily constrained.

**Questions:**

Are there application scenarios (e.g. data characteristics) where the constraint related to QRAM can be circumvented?

---

> ### Author Response · Authors · 2025-11-15
>
> We thank the reviewer for the detailed comments. Below we respond to each of your points in turn.
>
> 1, Regarding the remark that “it shares the $\sqrt{m}$ term with existing approaches …”, we feel this characterization is somewhat partial and does not capture the main contribution of our work. The primary advantage of our framework over prior quantum results is not in this restricted linear-regression case, but in its ability to handle much more general loss functions, such as $\ell_p$ regression and, in particular, the $\gamma_p$ family, including many important non-homogeneous losses. For these numerous, important, and technically challenging GLM cases, our quantum algorithms are (to the best of our knowledge) the first efficient ones, and the dependence on $m$ and $1/\varepsilon$ is essentially optimal up to logarithmic factors (see also our response to Reviewer QN5x).
>
> 2, Concerning the statement that “the result is an incremental advancement combining …”, we would like to clarify our perspective. In hindsight, once the final framework is laid out, it may look like a clean combination of known classical and quantum components. However, arriving at this unified picture is far from straightforward. Our work required carefully studying the most advanced and technically involved classical results and designing how to use existing quantum tools in a precise and efficient way so that the overall pipeline achieves genuine quantum speedups for a broad class of GLM losses. At the same time, we deliberately prioritized readability: we aimed to present the concrete structure of the quantum algorithms and the proof ideas as clearly as possible, so that the framework can serve as a transparent foundation for further applications of these techniques in the quantum algorithms community.
>
> 3, You mention that “...it unclear which results are prior work and which are novel …”. We respectfully disagree. When introducing Def. 5 and Def. 6, we explicitly state (see line 272) that these notions are taken from Jambulapati et al., and we do not intend to claim them as our own contributions. That said, we agree this attribution can be made more prominent, and in the revised version we will add explicit references directly in the statements of these definitions to avoid any ambiguity.
>
> 4, You also write that “the main manuscript provides no clear description …”. Due to space constraints in the main text, we made a conscious choice to present the core ideas and high-level structure of Algorithms 1 and 2 there, and to move many technical details to the appendix. In Appendix B, we provide detailed, concrete explanations and proofs for each subroutine, and in the main text we explicitly point the reader to these sections.
>
> Regarding ModLevApprox specifically: it is a slight modification of the quantum leverage score approximation routine and forms a basic primitive for Algorithm 1. We emphasize this in lines 244--247 of the main text. Conceptually, Algorithm 1 is a  iterative improvement procedure, where each iteration uses quantum leverage score approximation as a subroutine. The technical difficulty lies not in re-stating Apers and Gribling’s result, but in rigorously analyzing this iterative process to show that it indeed converges to the desired multiscale structure while preserving the overall quantum speedup.
>
> 5, Finally, you express concerns about the QRAM-based framework and ask whether there are application scenarios where the QRAM constraint can be circumvented. We would first like to stress that, in the theoretical quantum algorithms community, QRAM is a widely used and relatively standard assumption; many recent quantum algorithmic works rely on it. To give just a few examples from recent top ML venues:
>
> Quantum (Inspired) $D^2$-sampling with Applications (ICLR 2025), Near-Optimal Quantum Algorithms for Computing (Coarse) Correlated Equilibria of General-Sum Games (NeurIPS 2025), Quantum Speedup for Hypergraph Sparsification (ICML 2025), Quantum Algorithms for Finite-Horizon Markov Decision Processes (ICML 2025).
>
> At the same time, on the experimental side there has been encouraging progress toward physically realizing QRAM-like architectures, including (but not limited to):
>
> Quantum random access memory with transmon-controlled phonon routing(PhysRevLett.134.210601), Experimental realization of the bucket-brigade quantum random access memory(arxiv:2506.16682), Demonstrating coherent quantum routers for bucket-brigade QRAM on a superconducting processor(arxiv:2505.13958).
>
> These works suggest that QRAM is gradually moving from pure theory toward experimentally motivated architectures. In future work, we also plan to explore variants of our algorithms that  more implementation-friendly data-access models.
>
>
>
> We hope that these points address your concerns and clarify the scope and significance of our contributions, and that they may prompt a reconsideration of whether the current overall score fully reflects our work.

---

### Official Review · Reviewer_QN5x · 2025-11-01

**Soundness:** 4
**Presentation:** 4
**Contribution:** 3
**Rating:** 8
**Confidence:** 4

**Summary:**

This manuscript presents quantum algorithms for many regression tasks, including linear, multiple, Lasso, Ridge, Huber, $\ell_p$, and $\delta_p$ regressions. These algorithms are based on a unified framework, where the core quantum technique is a fast algorithm for approximating leverage scores due to Apres & Gribling (2024).  On top of this, a generalized linear model sparsifier is constructed using the techniques from Cohen & Peng (2015) and Jambulapati et al. (2024).

Based on this GLM sparsification framework, the authors derived a quantum algorithm for various regression problems. These quantum algorithms are faster than best-known classical algorithms in the not-too-precise regime, and are faster than previous quantum algorithms by Song et al. (2023) when the linear model is sparse.

**Strengths:**

The manuscript is very well written. The statements are supported with rigorous proofs. Although this work relies on existing quantum subroutines, it incorporates quantum subroutines in a nontrivial way to develop multiscale leverage score overestimation and quantum importance sampling, which are key components of this framework. Overall, I think this submission makes a solid theoretical contribution to solving regression problems.

**Weaknesses:**

For linear, multiple, and Ridge-regressions, the quantum algorithms proposed in this work are faster than previous quantum algorithms by Song et al. (2023), mainly due to the sparsity $r \leq n$. It looks like this improvement is merely due to the fact that the authors considered sparse regression models. Is it the case, or is the improvement truly a consequence of the new techniques?

**Questions:**

This is related to my question in the previous section. If we consider sparse regression models with sparsity $r$ and use the quantum algorithm by Song et al. (2023) to solve it, what would the dependence on $r$ look like?

A minor typographical issue in lines 111-112: repeated word "constructing".

---

> ### Author Response · Authors · 2025-11-15
>
> We thank you for the positive and constructive feedback.
>
> Regarding your main concern: in the special case of linear regression, our framework only requires standard leverage scores rather than the full multiscale leverage score (MLSO) machinery. Concretely, in Algorithm 2, Steps 2–4 can be replaced by the quantum leverage score routine of Apers and Gribling (AG23). Under this simplification, Algorithm 2 essentially reduces to a quantum spectral approximation procedure in the sense of Theorem 3.1 of AG23, and once a spectral sparsifier has been constructed, we can approximately solve linear regression and its simple variants (e.g., multiple and ridge regression) on top of that sparsifier using standard techniques.
>
> In hindsight, the linear regression result of Song et al. (2023) can also be viewed as a corollary of Theorem 3.1 of AG23, but with a coarse analysis that  replaces the row-sparsity parameter $r$ by $n$. What we would like to emphasize is that the main advantage of our work over prior results is not in this restricted linear case, but in the treatment of much more general loss functions, especially those not satisfying homogeneity, and in particular the $\gamma_p$ family. In these general GLM settings, one genuinely needs our MLSO and quantum importance sampling framework; the linear case is the simplest instance where these ingredients reduce to standard leverage scores.
>
> We have also become aware of concurrent work (arXiv:2509.16801), which studies quantum $\ell_p$ regression problems. Compared to this concurrent result, our quantum GLM sparsification achieves strictly better dependence on the accuracy parameter, scaling as $1/\varepsilon$ rather than $1/\varepsilon^3$. Moreover, when $m =O(n^2)$, the quantum lower bound for graph sparsification (Apers \& de Wolf, 2020) implies that our dependence on $1/\varepsilon$ in quantum GLM sparsification is optimal up to logarithmic factors.
>
> We very much appreciate your insightful comments and will update the manuscript to clarify these points and to fix the typographical error (the repeated word “constructing”) that you pointed out.

---

### Author Response · Authors · 2025-11-22

We are grateful to all reviewers for their careful reading of our work and for their constructive comments.  We are encouraged that they found our quantum GLM sparsification framework for many regression problems conceptually clear and broadly useful  (QN5x, C66o, aYYA), our technical development and proofs solid (QN5x, aYYA), and our sparsifier complexity for linear regression and its variants in the sparse regime noteworthy (Ewfu).

In addition to fixing several typos, we have addressed the main concerns raised in the reviews. In particular, we have expanded the discussion of Hamoudi's “preparing many copies of a quantum state’’ technique, clarified the relationship between our quantum GLM sparsification framework and linear regression, and added an explicit discussion of the relevant quantum lower bounds. We have also updated and refined several citations and noted a concurrent and independent work (arXiv:2509.16801). All revisions are marked in red in the revised manuscript.

---

### Note · Authors · 2026-01-28

I have read and agree with the venue's withdrawal policy on behalf of myself and my co-authors.

---

### Meta-Review · Area_Chair_FRQn · 2025-12-10

**Summary:**

This paper proposes a unified quantum framework for sparsification in generalized linear models, extending recent results by Jambulapati et al. (2024) in the classical setting using the quantum leverage score approximation technique by Apers and Gribling (2024). The framework encompasses a wide range of regression problems, including linear, Ridge, Lasso, and Huber regression, and achieves improvements over both classical algorithms and prior quantum algorithms.

Although reviewers generally found the paper technically solid, there was significant concern about the level of novelty, with several reviewers noting that the main contributions appeared to be a combination of the recent sparsification techniques by Jambulapati et al. (2024), the quantum leverage score approximation technique by Apers and Gribling (2024), an approach by Li et. al. (2019) to estimate the correct normalization constant, and a quantum algorithm by Hamoudi (2022) to pre-emptively sample before the correct normalization constant is determined, by preparing multiple copies. Finally, another weakness of the paper is the assumption of the QRAM model, which despite previous works in ML venues, nevertheless restricts its practical applicability.

**Reviewer Concerns:**

I believe the questions by Reviewer C66o regarding the source of the quantum speedup were addressed in the rebuttal. However, the concerns raised by other reviewers about the paper’s technical novelty and reliance on the QRAM model, while acknowledged and partially discussed, were not fully resolved.

**Reviewer Scores:**

After the initial rebuttal, Reviewer aYYA raised their score from 2 to 4. Given the above points about concerns about technical novelty and the QRAM model, it is not clear to me that other reviewers would have raised their scores.

---

### Decision · Program_Chairs · 2026-01-26

Reject